# P$^2$C$^2$Net: PDE-Preserved Coarse Correction Network for Efficient Prediction of Spatiotemporal Dynamics

**Qi Wang[1], Pu Ren[2], Hao Zhou[1], Xin-Yang Liu[3], Zhiwen Deng[4], Yi Zhang[4], Ruizhi Chengze[4], Hongsheng Liu[4], Zidong Wang[4], Jian-Xun Wang[3], Ji-Rong Wen[1], Hao Sun[1,*], Yang Liu[5,*]**

[1]Gaoling School of Artificial Intelligence, Renmin University of China, Beijing, China
[2]Department of Civil and Environmental Engineering, Northeastern University, Boston, MA, USA
[3]Department of Aerospace and Mechanical Engineering, University of Notre Dame, Notre Dame, IN, USA
[4]Huawei Technologies, Shenzhen, China
[5]School of Engineering Science, University of Chinese Academy of Sciences, Beijing, China

Emails: `qi_wang@ruc.edu.cn` (Q.W.); `haosun@ruc.edu.cn` (H.S.); `liuyang22@ucas.ac.cn` (Y.L.)

## Abstract

When solving partial differential equations (PDEs), classical numerical methods often require fine mesh grids and small time stepping to meet stability, consistency, and convergence conditions, leading to high computational cost. Recently, machine learning has been increasingly utilized to solve PDE problems, but they often encounter challenges related to interpretability, generalizability, and strong dependency on rich labeled data. Hence, we introduce a new PDE-Preserved Coarse Correction Network (P$^2$C$^2$Net) to efficiently solve spatiotemporal PDE problems on coarse mesh grids in small data regimes. The model consists of two synergistic modules: (1) a trainable PDE block that learns to update the coarse solution (i.e., the system state), based on a high-order numerical scheme with boundary condition encoding, and (2) a neural network block that consistently corrects the solution on the fly. In particular, we propose a learnable symmetric Conv filter, with weights shared over the entire model, to accurately estimate the spatial derivatives of PDE based on the neural-corrected system state. The resulting physics-encoded model is capable of handling limited training data (e.g., 3–5 trajectories) and accelerates the prediction of PDE solutions on coarse spatiotemporal grids while maintaining a high accuracy. P$^2$C$^2$Net achieves consistent state-of-the-art performance with over 50% gain (e.g., in terms of relative prediction error) across four datasets covering complex reaction-diffusion processes and turbulent flows.

## 1 Introduction

Complex spatiotemporal dynamical systems are pivotal and commonly seen in numerous fields such biology, meteorology, fluid mechanics, etc. The behaviors of these systems are primarily governed by partial differential equations (PDEs), conventionally solved by numerical methods [1–4]. However, direct numerical simulation (DNS) demands in-depth knowledge of the underlying physics, and the efficacy of numerical solutions is intricately linked to the resolution of mesh grids and time steps. High-resolution spatiotemporal grids are essential for accurate and convergent calculations, yet leading to substantial computational costs. For instance, simulating the flow field around a large aircraft [5, 6] involves creating over millions of mesh nodes and may consume vast simulation time even on high performance computing. Additionally, any changes in initial and boundary conditions (I/BCs) or design parameters necessitate recalculations, further compounding the complexity.

---

*Corresponding authors

Recently, tremendous efforts have been placed on machine learning for data-driven simulation of these systems [7–9], demonstrating promising potential. These methods do not require the *a priori* knowledge of physics and, meanwhile, help bypass some traditional constraints, e.g., the smallest size of mesh grid and time step to guarantee solution accuracy, stability and convergence [10]. However, they typically face issues of poor interpretability, weak generalizability and strong dependency of rich labeled data. Their performance deteriorate significantly particularly in small data regimes.

Embedding prior physics knowledge into the learning process has demonstrated effective to overcome the aforementioned issues. A brute-force way lies in creating regularizers (e.g., the residual form of PDEs and I/BCs) as "soft" penalty in the loss function, e.g., the family of physics-informed neural works (PINNs) [11–17]. However, such a strategy has limited scalability and generalizability, and the solution accuracy relies largely on a proper selection of loss weight hyperparameters. Embedding physics explicitly into the network architecture, which imposes "hard" constraints such as physics-encoded recurrent convolutional neural network (PeRCNN) [18, 19], possesses better generalizability as well as offers better convergence and flexibility for model training without the need of fine-tuning hyperparameters. Nevertheless, existing methods fail to handle coarse mesh grids and suffer from instability issues especially for long-range prediction of dynamics. Hybridizing classical numerical schemes and neural networks, e.g., the learned interpolation (LI) model [20], can enable accelerated simulation on coarse mesh grids with satisfied accuracy. Yet, since the numerical part is non-trainable, such models still require rich labeled data to retain accuracy.

To tackle these critical challenges, we introduce the $P^2C^2$Net model for efficient prediction of spatiotemporal dynamics on coarse mesh grids in small training data regimes. Specifically, a trainable PDE block (a white box) is designed to learn the coarse solution at low resolution, where the temporal marching of system states is handled by a fourth-order Runge-Kutta (RK4) scheme. We also propose a learnable symmetric Conv filter for more accurate estimation of spatial derivatives on coarse grids, as required in PDE block. A neural network (NN) block, which serves as a correction module, is further introduced to correct the coarse solution, restoring information lost due to reduced resolution. We also encode BC into the solution via a padding strategy. Our primary contributions are threefold:

- We propose a new physics-encoded correction learning model ($P^2C^2$Net) to efficiently predict complex spatiotemporal dynamics on coarse mesh grids. The model requires only a small set of training data and possesses plausible generalizability.

- We introduce a structured Conv filter that preserves symmetry to improve the estimation accuracy of coarse spatial derivatives required in the solution updating process, which makes the PDE block trainable with flexibility of handling coarse grids.

- $P^2C^2$Net achieves consistent state-of-the-art performance with at least 50% gain (e.g., in terms of relative prediction error) across four datasets covering complex reaction-diffusion (RD) processes and turbulent flows, simultaneously retaining accuracy and efficiency.

## 2  Related work

**Numerical methods.**  Conventionally, PDE systems are solved by classical numerical methods such as finite difference [1], finite volume [2], finite element [3], and spectral methods [4]. These methods often require fine mesh grids and reasonable time stepping to meet stability, consistency and convergence conditions. When dealing with large scale simulations or inverse analyses, the computational costs remain remarkably high.

**Deep learning methods.**  Given sufficient labeled training data, deep learning has been recently applied to solve PDE problems. Representative approaches include Conv-based NN models [9, 21], U-Net [22], ResNet [23], graph neural networks [24, 25], and Transformer-based models [26–28]. In addition, neural operators such as DeepONet [7], multiwavelet-based model (MWT) [29], Fourier neural operator (FNO) [8], and their variants [30–32] have been designed to directly learn mappings between function spaces, making them particularly well-suited for modeling PDE systems. Diffusion models have also been employed for prediction of spatiotemporal dynamics [33].

**Physics-aware learning methods.**  Recently, physics-aware deep learning has demonstrated great potential in modeling spatiotemporal dynamics under conditions of small training data. This paradigm can be divided into two categories based on the way of embedding PDE information: (1) *physics-informed*, and (2) *physics-encoded*. The former formalizes PDE soft constraints including equations

and I/BCs via loss regularization on point-wise or mesh-based NNs (e.g., PINN [11–13, 17], Phy-GeoNet [34], PhyCRNet [35], PhySR [36], etc.), while the latter imposes hard constraints via encoding PDE structures (e.g., equations, I/BCs, law of thermodynamics, symmetry) into NN architectures such as PeRCNN [18, 19], TiGNN [37], and EquNN [38]. Other related works include the PDE-Net models [39, 40] with designed convolution kernels approximating differential operators, thereby modeling the dynamics of the system.

**Hybrid learning methods.** The hybrid learning scheme represents a novel research direction that has emerged in recent years, which integrates numerical methods with NNs. The resulting solver can leverage a variety of classical numerical methods such as finite difference (e.g., PPNN [41], numerical discretization learning [42]), finite volume (e.g., LI [20] and TSM [43], and spectral methods (e.g., machine-learning-augmented spectral solver [44]). These approaches can operate on coarse grids, enabling faster simulations compared with traditional numerical solvers while retaining accuracy. However, since the numerical part is non-trainable, such models generally require rich labeled data.

## 3 Methodology

### 3.1 Problem formulation

Let us consider a spatiotemporal dynamical system governed by the general form of PDEs:

$$\frac{\partial \mathbf{u}}{\partial t} - \mathcal{F}(\mathbf{u}, \mathbf{u}^2, \cdots, \boldsymbol{\nabla}\mathbf{u}, \boldsymbol{\nabla}^2\mathbf{u}, \cdots; \boldsymbol{\mu}) = \mathbf{f}, \tag{1}$$

where $\mathbf{u}(\mathbf{x}, t) \in \mathbb{R}^n$ denotes the $n$-dimensional physical state within spatiotemporal domain $\Omega \times [0, T]$; $\partial\mathbf{u}/\partial t$ the first-order time derivative; $\mathcal{F}$ a linear/nonlinear function; $\boldsymbol{\nabla} \in \mathbb{R}^n$ the Nabla operator; $\boldsymbol{\mu}$ the PDE parameters (e.g., the Reynolds number $Re$); $\mathbf{f}$ the external force (e.g., $\mathbf{f} = \mathbf{0}$ for source-free cases). Besides that, we define the initial condition (IC) as $\mathcal{I}(\mathbf{u}, \mathbf{u}_t; \mathbf{x} \in \Omega, t = 0) = \mathbf{0}$ and the boundary condition (BC) as $\mathcal{B}(\mathbf{u}, \boldsymbol{\nabla}\mathbf{u}, \cdots; \mathbf{x} \in \partial\Omega) = \mathbf{0}$, where $\partial\Omega$ denotes the boundary of $\Omega$.

Our aim is to develop a *learnable coarse model* that accelerates the simulation and prediction of spatiotemporal dynamics based on a minimal set of sparse data (e.g., low-resolution data down-sampled across space and time). The learned model is expected to achieve high solution accuracy and superior generalizability over various PDE scenarios, including ICs, force terms, and PDE parameters.

### 3.2 Network architecture

Herein, we introduce the P$^2$C$^2$Net architecture, as shown in Figure 1, taking the simulation of Navier-Stokes (NS) flows as an example. The model is composed of four blocks, namely, the state variable correction block, the learnable PDE block, the Poisson block, and the NN block. Note that the network architecture is flexible and features a Poisson block that solves for the pressure term $p$, which is absent in other cases.

#### 3.2.1 The flow of data

In Figure 1 (**a**), the network architecture includes two paths: the upper path computes the coarse solution using a learnable PDE block, while the lower path is incorporated into the network to correct the solution on a coarse grid with a Poisson block and a NN block. The data flow operates as follows: (1) the network accepts $\mathbf{u}_k$ as input and processes it by the PDE block on the upper path, where the PDE block computes the residual of the governing equation. A filter bank, defined as a learnable filter with symmetry constraints, calculates the derivative terms based on the corrected solution (produced by the correction block). These terms are combined into an algebraic equation (a learnable form of $\mathcal{F}$). This process is incorporated into the RK4 integrator for solution update. (2) In the lower path, $\mathbf{u}_k$ is first corrected by the correction block, and $p_k$ is computed by the Poisson block. Inputs, including solution states $\{\mathbf{u}_k, p_k\}$ and their derivative terms, forcing term, and Reynolds number, are fed into the NN block. The output from this block serves as a correction for the upper path. (3) The final result $\mathbf{u}_{k+1}$ is obtained by combining the outputs from both the upper and lower paths. During the gradient back-propagation process, the NN block learns to correct the coarse solution output of the PDE block on the fly, and ensures that their combined results more closely approximate the ground truth solution.

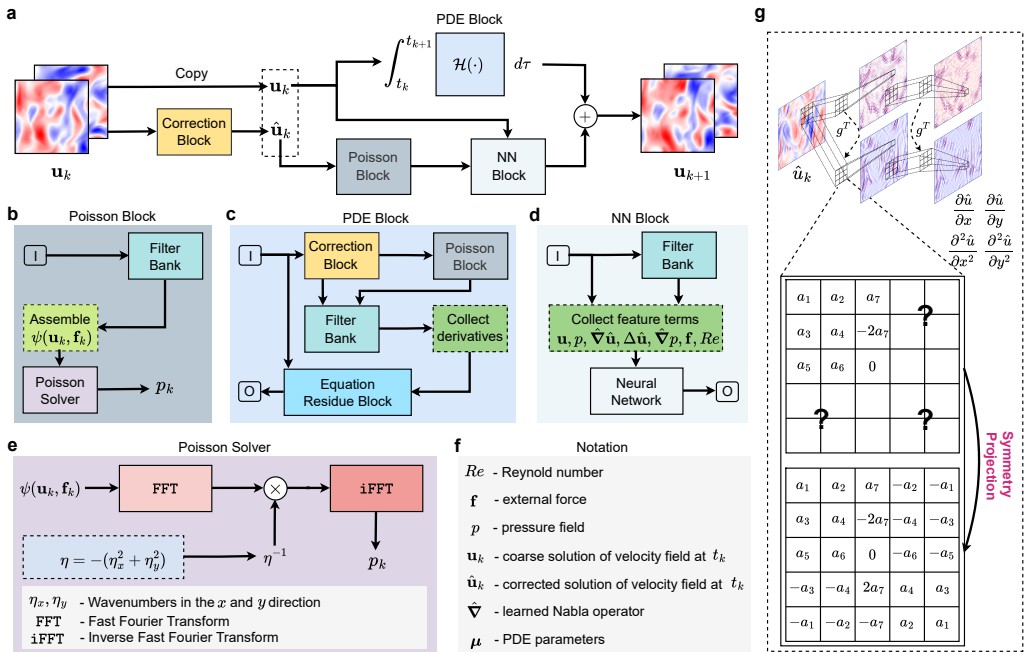

Figure 1: Schematic of P²C²Net for learning Navier-Stokes flows. (**a**), Overall model architecture. (**b**), Poisson block. (**c**), learnable PDE block. (**d**), NN block. (**e**), Poisson solver. (**f**), Symbol notations. (**g**), Conv filter with symmetric constraint.

### 3.2.2 RK4 integration scheme

Similar to standard numerical solvers, the goal is to predict the solution at every time step satisfying the underlying PDEs given specific I/BCs. Here, we aim to address the challenge of spatiotemporal dynamics evolution on coarse grids (e.g., low resolution). Given the coarse solution at timestep $t_k$, denoted by $\mathbf{u}_k$, we expect the model yielding an accurate prediction of $\mathbf{u}_{k+1}$, *v.i.z.*,

$$\mathbf{u}_{k+1} = \mathbf{u}_k + \int_{t_k}^{t_{k+1}} \left[ \mathcal{H}\left(\mathbf{u}(\tilde{\mathbf{x}}, \tau), \mathbf{u}^2(\tilde{\mathbf{x}}, \tau), \cdots, \boldsymbol{\nabla}\mathbf{u}(\tilde{\mathbf{x}}, \tau), \boldsymbol{\nabla}^2\mathbf{u}(\tilde{\mathbf{x}}, \tau), \cdots; \boldsymbol{\mu}\right) + \mathbf{f}(\tau) \right] d\tau \quad (2)$$

where $\mathcal{H}$ (the PDE block) denotes a learnable form of $\mathcal{F}$, which is discussed in Section 3.2.3; $\tilde{\mathbf{x}}$ represents the coarse grid coordinates. We herein employ the RK4 scheme as the integrator for time marching of the dynamics, offering the fourth-order accuracy ($\mathcal{O}(\delta t^4)$), where $\delta t = t_{k+1} - t_k$ is the coarse time step. More details on RK4 can be found in Appendix Section B.3.

### 3.2.3 Learnable PDE Block

We assume that the PDE formulation in Eq. (1) is given, e.g., the explicit expression of $\mathcal{F}$ is known. With coarse mesh grids and large time stepping, numerical methods such as finite difference (FD) tend to diverge. This issue becomes more pronounced with greater grid coarsening. Hence, we propose a learnable PDE block (depicted in Figure 1(**c**)), denoted by $\mathcal{H}$, to approximate $\mathcal{F}$ on coarse grids, so as to enable the coarse simulation simultaneously retaining efficiency and accuracy, expressed as

$$\mathcal{H}\left(\mathbf{u}_k, \mathbf{u}_k^2, \cdots, \boldsymbol{\nabla}\mathbf{u}_k, \boldsymbol{\nabla}^2\mathbf{u}_k, \cdots; \boldsymbol{\mu}\right) \leftarrow \mathcal{F}\left(\mathbf{u}_k, \mathbf{u}_k^2, \cdots, \hat{\boldsymbol{\nabla}}\hat{\mathbf{u}}_k, \hat{\boldsymbol{\nabla}}^2\hat{\mathbf{u}}_k, \cdots; \boldsymbol{\mu}\right) \quad (3)$$

where $\mathbf{u}_k$ denotes the coarse solution at time $t_k$; $\hat{\mathbf{u}}_k$ the corresponding neural-corrected coarse solution state, obtained by the correction block shown in Figure 1(**a**) and (**c**), used for estimation of spatial derivatives, e.g., $\hat{\mathbf{u}}_k = \text{NN}(\mathbf{u}_k)$. Here, $\hat{\boldsymbol{\nabla}}$ represents a learnable Nabla operator composed of a Conv filter with symmetric constraint (e.g., an enhanced FD kernel for numerical approximation of spatial derivatives). Through the RK4 integration, we are then able to predict the coarse solution for the next time step. Note that the PDE parameters $\boldsymbol{\mu}$ can be set as trainable if unknown. Despite information loss at the low resolution, such a learnable PDE block allows for more accurate derivative estimation on coarse grids and ensures better adherence of the updated coarse solution to underlying PDEs. Clearly, such a block adds a fully interpretable "white box" to the overall network architecture.

**Correction block:** The correction block takes a NN to correct the coarse solution. In particular, we choose FNO [8] as the neural correction operator performed on the coarse solution field in such a block. In the Fourier space, FNO decomposes the input data into components with specific frequencies, processes each component separately, and then applies Fourier transform to restore the updated spectral features back to the physical domain. The layer-wise update can be expressed as:

$$\mathbf{v}^{l+1}(\tilde{\mathbf{x}}) = \sigma\left(\mathbf{W}^l\mathbf{v}^l(\tilde{\mathbf{x}}) + \left(\mathcal{K}(\phi)\mathbf{v}^l\right)(\tilde{\mathbf{x}})\right) \tag{4}$$

where, $\mathbf{v}^l(\tilde{\mathbf{x}})$ denotes the $l$-th layer latent feature map on the coarse grids $\tilde{\mathbf{x}}$. Note that $\mathbf{v}^0(\tilde{\mathbf{x}}) = \mathcal{P}(\mathbf{u}_k)$, in which $\mathcal{P}$ is a local transformation function that lifts $\mathbf{u}_k$ to a higher dimensional representation. Here, $\mathcal{K}(\phi)(\mathbf{z}) = \texttt{iFFT}(\mathbf{R}_\phi \cdot \texttt{FFT}(\mathbf{z}))$ denotes a kernel integral transformation of the latent feature map $\mathbf{z}$, which encompasses Fourier transform, spectral filtering and convolution in the frequency domain $\mathbf{R}_\phi$, and inverse Fourier transform; $\phi$ the network parameters; $\sigma(\cdot)$ the GELU activation function; $\mathbf{W}^l$ the weights of a linear transformation. Given an $L$-layer FNO, the corrected coarse solution reads $\hat{\mathbf{u}}_k = \mathcal{Q}\left(\mathbf{v}^L(\tilde{\mathbf{x}})\right)$, where $\mathcal{Q}$ is a local projection function. Details of the FNO correction block are found in Appendix Section B.1.

**Conv filter with symmetric constraint:** Based on FD schemes, spatial derivatives can be approximated by central difference stencils represented by convolution kernels [35, 39–41]. Such an approach often requires fine mesh grids to ensure accuracy; otherwise on coarse grids, there exists solution divergence issue. To this end, we leverage our understanding of FD stencils and propose a Conv filter with symmetric constraints to improve the accuracy of spatial derivative approximation on coarse mesh grids. As shown in Figure 1(**g**), the filter weights are transformed into a $5 \times 5$ matrix **g** with 7 learnable parameters (Table S1 demonstrates the significant differences in results across various kernel sizes when applied to the Burgers dataset.), which satisfies symmetry, to estimate the first-order derivative along the horizontal direction (e.g., the vertical direction takes $\mathbf{g}^T$), *v.i.z.*,

$$\mathbf{g} = \begin{pmatrix} a_1 & a_4 & a_7 & -a_4 & -a_1 \\ a_2 & a_5 & -2a_7 & -a_5 & -a_2 \\ a_3 & a_6 & 0 & -a_6 & -a_3 \\ -a_2 & -a_5 & 2a_7 & a_5 & a_2 \\ -a_1 & -a_4 & -a_7 & a_4 & a_1 \end{pmatrix} \tag{5}$$

Although the number of learnable parameters in the Conv kernel is limited, the coarse derivatives can still be accurately approximated after the model is trained (see the ablation study in Section 4.2). The structured filter is designed for Conv operation which satisfies the Order of Sum Rules stated in Lemma 1 [39] (see Appendix Section A for more details).

**Lemma 1:** *A 2D filter $\mathbf{g}_{m \times m}$ has sum rules of order $\iota = (\iota_1, \iota_2)$, where $\iota \in \mathbb{Z}_+^2$, provided that*

$$\sum_{k_1=-\frac{m-1}{2}}^{\frac{m-1}{2}} \sum_{k_2=-\frac{m-1}{2}}^{\frac{m-1}{2}} k_1^{\alpha_1} k_2^{\alpha_2} g[k_1, k_2] = 0 \tag{6}$$

*for all $\alpha = (\alpha_1, \alpha_2) \in \mathbb{Z}_+^2$ with $|\alpha| := \alpha_1 + \alpha_2 < |\iota|$ and for all $\alpha \in \mathbb{Z}_+^2$ with $|\alpha| = |\iota|$ but $\alpha \neq \iota$. If this holds for all $\alpha \in \mathbb{Z}_+^2$ with $|\alpha| < A$ except for $\alpha \neq \check{\alpha}$ with certain $\check{\alpha} \in \mathbb{Z}_+^2$ and $|\alpha| = B < A$, then we say g to have total sum rules of order $A \setminus \{B + 1\}$.*

**Corollary 1:** *The filter $\mathbf{g}$ we designed in Eq. (5) has the sum rules of order (1, 0). By adjusting the parameters in $\mathbf{g}$, e.g., $a_7 \to 0$, $a_6 + 8a_5 \to 0$, it satisfies the total sum rules of order $5 \setminus \{2\}$, and achieves an approximation to the first-order derivative with the fourth-order accuracy. For example, for a smooth function $\omega(x, y)$ and small perturbation $\varepsilon > 0$, we have [39]:*

$$\frac{1}{\varepsilon} \sum_{k_1=-\frac{m-1}{2}}^{\frac{m-1}{2}} \sum_{k_2=-\frac{m-1}{2}}^{\frac{m-1}{2}} g[k_1, k_2]\omega(x + \varepsilon k_1, y + \varepsilon k_2) = C\frac{\partial \omega(x, y)}{\partial x} + \mathcal{O}(\varepsilon^4), \text{ as } \varepsilon \to 0. \tag{7}$$

The proof of Corollary 1 can be found in Appendix Section A. Given that the order of sum rules is closely related to the order of vanishing moments in the wavelet theory, Lemma 1 ensures that the filter not only maintains high sensitivity to local changes in the feature but also effectively suppresses irrelevant low-frequency components, thereby enhancing the estimation accuracy of the first derivative [39]. Furthermore, Corollary 1 further guarantees that the designed filter can approximate the first-order derivative with up to the fourth-order accuracy by properly adjusting the trainable parameters.

**BC encoding:** To ensure that the predicted solution complies with the given BCs, while also retaining the shape of the feature map after Conv operations, we employ a BC hard encoding method [19]. In particular, we consider periodic BCs in this work and apply padding padding, as shown in Figure 2, on the predicted solutions. Such an encoding technique not only ensures the compliance of the predicted solution with the BCs, but also enhances the solution's accuracy.

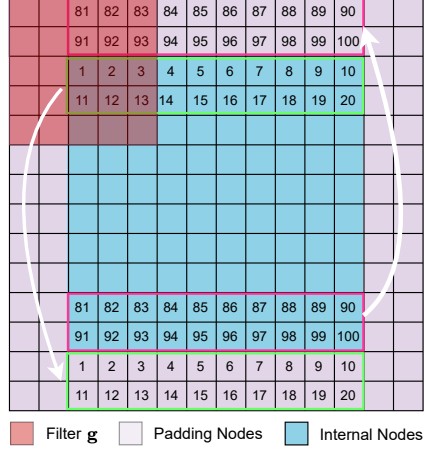

Figure 2: Periodic BC padding.

### 3.2.4 Poisson block

When solving NS equations, there exists the pressure term $p$ in the governing PDEs. However, for incompressible flows, the pressure can be inferred by solving a Poisson equation. We designed this block to achieve this aim, the poisson equation namely, $\Delta p = \psi(\mathbf{u}, \mathbf{f})$, where $\psi(\mathbf{u}, \mathbf{f}) = -2\left(u_y v_x - u_x v_y\right) + \nabla \cdot \mathbf{f}$ for 2D problems and the subscripts denote spatial differentiation. Hence, we leverage the spectral method to estimate the pressure quantity (see Appendix Section B.2), as depicted in Figure 1(**e**), which updates $p_k$ based on $\psi(\mathbf{u}_k, \mathbf{f}_k)$. The resulting Poisson block (Figure 1(**b**)) efficiently derives the pressure field from the velocity field and external force on the fly, streamlining computations without the need for labeled training data of pressure.

### 3.2.5 NN block

It is noted that the predictions by the learnable PDE block on coarse grids may encounter instability issue due to error accumulation over time, especially in scenarios involving long-range rollout prediction. Hence, we propose an additional NN block to consistently correct the coarse solution predicted by the PDE block on the fly, restoring information lost due to reduced resolution. This module can be any trainable NN model, such as FNO [8], DenseCNN [41], or UNet [13, 45]. In particular, we consider FNO as the NN block, with more details found in Appendix Section B.1. However, the NN block is not always necessary. For cases of simpler systems, such as the Burgers equation, the learnable PDE block alone can perform well where the NN block can be omitted.

### 3.2.6 Model generalization

We herein introduce the setup of the model's generalizability over various PDE scenarios, including ICs, force terms, and PDE parameters (e.g., the Reynolds number $Re$). The time marching in our network design inherently integrates ICs, automatically ensuring generalization over ICs given a well trained model. We embed the force term in the learnable PDE Block (naturally in the PDE formulation) shown in Figure 1(**c**), and meanwhile incorporate it as a feature map into the NN Block as part of the input (see Figure 1(**c**)). This enables the joint learning of force feature variations for generalization. In addition, the Reynolds number ($Re$) is incorporated by creating a 2D feature map embedding (e.g., for the 2D case) by introducing two trainable vectors $\mathbf{a}$ and $\mathbf{b}$, namely, $Re_{\text{embb}} = 1/Re \cdot (\mathbf{a} \otimes \mathbf{b})$, in both the PDE Block and the the NN Block. Such an approach can correct the error propagation of the diffusion term in the PDE Block caused by coarse grids, and enhance the model's ability of generalization across different $Re$'s.

## 4   Experiments

We evaluate the performance of $\text{P}^2\text{C}^2\text{Net}$ against several baseline models across diverse PDE systems, including fluid flows and RD processes. The results have demonstrated the superiority of our model in terms of solution accuracy and generalizability thanks to the unique design of embedding PDEs into the network. The source codes and data are found at https://github.com/intell-sci-comput/P2C2Net (PyTorch) and https://gitee.com/intell-sci-comput/P2C2Net.git (MindSpore).

Table 1: Summary of datasets and training implementations. Note that "→" denotes the downsampling process from the original resolution (simulation) to the low resolution (training).

| Dataset | Numerical Method | Spatial Grid | Time Steps (Temporal Grid) | # of Training Trajectories | # of Testing Trajectories | Rollout Steps |
|---|---|---|---|---|---|---|
| Burgers | FD | $100^2 \rightarrow 25^2$ | 400 | 5 | 10 | 20 |
| FN | FD | $128^2 \rightarrow 64^2$ | $5500 \rightarrow 1375$ | 3 | 10 | 32 |
| GS | FD | $128^2 \rightarrow 32^2$ | $4000 \rightarrow 1000$ | 3 | 10 | 50 |
| NS | FV | $2048^2 \rightarrow 64^2$ | $153600 \rightarrow 4800$ | 5 | 10 | 32 |

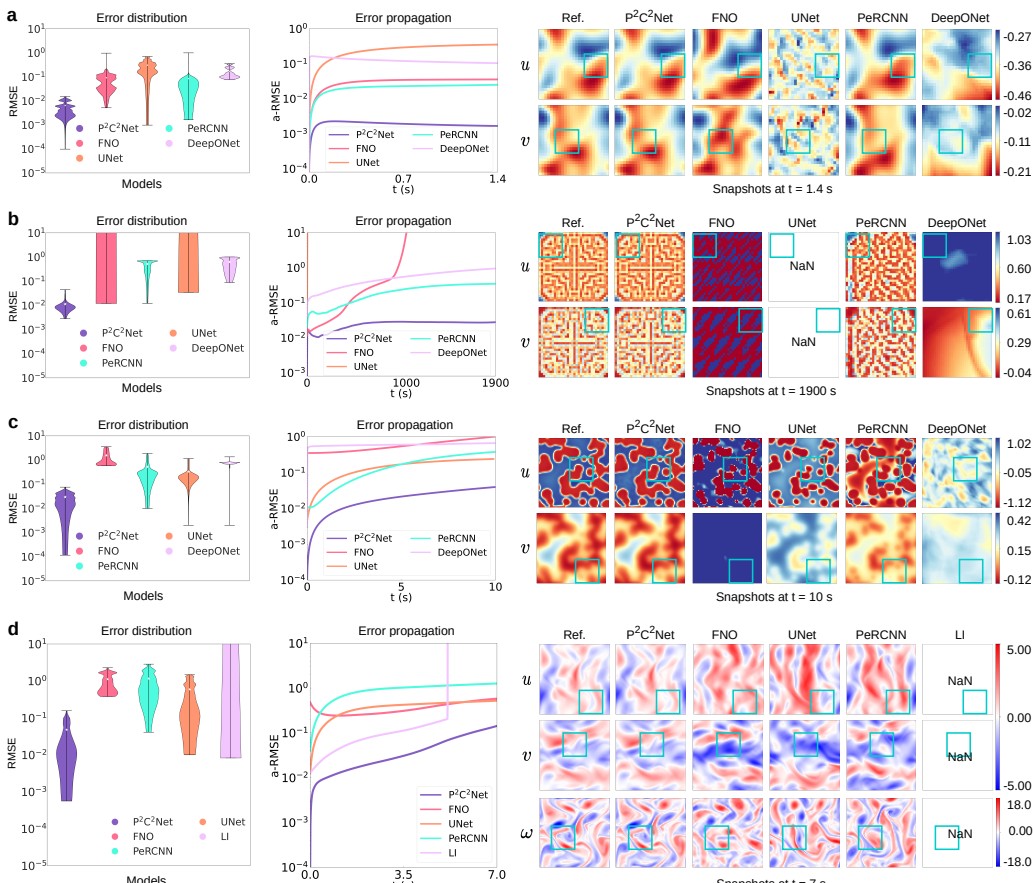

Figure 3: An overview of the comparison between our $P^2C^2$Net and baseline models, including error distributions (left), error propagation curves (middle), and predicted solutions (right). (**a**)-(**d**) show the qualitative results on Burgers, GS, FN, and NS equations, respectively. These PDE systems are trained with grid sizes of $25 \times 25$, $32 \times 32$, $64 \times 64$, and $64 \times 64$ accordingly.

## 4.1 Setup

**Datasets.** We consider four PDE datasets, including 2D Burgers, FitzHugh-Nagumo (FN), Gray-Scott (GS), and NS equations. Each dataset exhibits intricate spatiotemporal patterns, presenting significant challenges for prediction on coarse grids. High-order FD and finite volume (FV) methods are utilized to generate these datasets. We segment the training samples based on different rollout lengths. Additionally, the simulated high-resolution data $\mathbf{u} \in \mathbb{R}^{n_t \times n \times H \times W}$ is down-sampled across space and time to low-resolution counterparts $\tilde{\mathbf{u}} \in \mathbb{R}^{n'_t \times n \times H' \times W'}$ where $n'_t < n_t$, $H' < H$, $W' < W$. The low-resolution data serves as labels for training. Moreover, for each PDE case, only 3~5 sets of data trajectories are used for training while 10 trajectories are applied to test the model performance. The summary of datasets and training configuration is shown in Table 1. More details regarding datasets and their simulations can be found in the Appendix (see Table S2 and Section E).

**Evaluation metrics.** Four metrics are used to assess the model's performance: Root Mean Square Error (RMSE), Mean Absolute Error (MAE), Mean Normalized Absolute Difference (MNAD), and High Correction Time (HCT). The definition of each metric is listed in Appendix Section D.

**Model training.** Given the low-resolution training data, we aim to learn the evolution of spatiotemporal dynamics on coarse grids with bigger time stepping. This learning task is formulated as an auto-regressive rollout problem, where the models are only constrained by a low-resolution data loss. The loss function is defined as $\mathcal{J}(\boldsymbol{\theta}) = \frac{1}{BN} \sum_{i=1}^{B} \sum_{j=1}^{N} MSE\left(\check{\mathcal{S}}_{ij}, \mathcal{S}_{ij}\right)$, where $\check{\mathcal{S}}_{ij}$ denotes the rollout-predicted coarse solution for the $j$-th sample in the $i$-th batch, $\mathcal{S}_{ij}$ the corresponding label, $B$ and $N$ the number of batches and the batch size, and $\boldsymbol{\theta}$ the trainable parameters. The detailed network parameters and training configurations are provided in Appendix Section B.

**Baseline models.** To validate the superiority of the proposed P$^2$C$^2$Net model, we conducted comparisons with various baseline models, including FNO [8], UNet [45], PeRCNN [19], DeepONet [7], and Learned Interpolation (LI) [20]. The descriptions and training setup of the baseline models are provided in Appendix Section C.

### 4.2 Main results

Figure 3 presents the results of comparison between P$^2$C$^2$Net and baselines, including error distribution, error propagation, and predicted trajectories. Moreover, Table 2 provides the quantitative model performance results.

**Burgers Equation.** Generally, our P$^2$C$^2$Net and baseline models (PeRCNN and FNO) all capture the evolving dynamics and provide acceptable results as shown in the right part of Figure 3(**a**). However, our method shows a notably lower error level compared with baselines, as emphasized in the error distribution and error propagation presented in the left and middle parts of Figure 3(**a**). Moreover, Table 2 substantiates this observation, revealing a significant improvement in our model's performance compared with the baseline models from a minimum of 93.4% to a maximum of 211.7%. We further conducted boundary condition generalization tests, as detailed in Appendix Section G.1.

Table 2: Quantitative results of our model and baselines. For the case of Burgers, GS, and FN, our model inferred the test set's upper time limits of 1.4 s, 2000 s, and 10 s, respectively, as the trajectories of dynamics get stabilized. We take these limits in HCT to facilitate evaluation metrics calculation.

| Case | Model | RMSE | MAE | MNAD | HCT (s) |
|---|---|---|---|---|---|
| Burgers | FNO | 0.0980 | 0.0762 | 0.062 | 0.3000 |
|  | UNet | 0.3316 | 0.2942 | 0.2556 | 0.0990 |
|  | DeepONet | 0.2522 | 0.2106 | 0.1692 | 0.0020 |
|  | PeRCNN | 0.0967 | 0.1828 | 0.1875 | 0.4492 |
|  | P$^2$C$^2$Net (Ours) | **0.0064** | **0.0046** | **0.0037** | **1.4000** |
|  | Promotion (↑) | 93.4% | 94.0% | 94.0% | 211.7% |
| GS | FNO | NaN | NaN | NaN | 354 |
|  | UNet | NaN | NaN | NaN | 4 |
|  | DeepONet | 0.3921 | 0.2670 | 0.2670 | 852 |
|  | PeRCNN | 0.1586 | 0.0977 | 0.0976 | 954 |
|  | P$^2$C$^2$Net (Ours) | **0.0135** | **0.0062** | **0.0062** | **2000.0** |
|  | Promotion (↑) | 91.5% | 93.7% | 93.6% | 109.6% |
| FN | FNO | 0.8935 | 0.5447 | 0.2593 | 3.5000 |
|  | UNet | 0.1730 | 0.0988 | 0.0470 | 6.5000 |
|  | DeepONet | 0.5474 | 0.3737 | 0.1779 | 0.5128 |
|  | PeRCNN | 0.5703 | 0.2258 | 0.1075 | 5.3750 |
|  | P$^2$C$^2$Net (Ours) | **0.0390** | **0.0149** | **0.0071** | **10.000** |
|  | Promotion (↑) | 77.5% | 84.9% | 84.9% | 53.8% |
| NS | FNO | 1.0100 | 0.7319 | 0.0887 | 2.5749 |
|  | UNet | 0.8224 | 0.5209 | 0.0627 | 3.9627 |
|  | LI | NaN | NaN | NaN | 3.5000 |
|  | PeRCNN | 1.2654 | 0.9787 | 0.1192 | 0.6030 |
|  | P$^2$C$^2$Net (Ours) | **0.3533** | **0.1993** | **0.0235** | **7.1969** |
|  | Promotion (↑) | 57.0% | 61.7% | 62.5% | 81.6% |

**GS Equation.** The primary challenge of this dataset lies in its sparsity and the intricate patterns it presents, as depicted in Figure 3(**b**) (right). Each baseline model struggles to accurately predict the trajectories, especially the UNet model demonstrating significant divergence. Nevertheless, our method exhibits superior stability and is capable of learning the underlying dynamics. This is further validated by the error analysis presented in Figure 3(**b**), where our P$^2$C$^2$Net model outperforms the baselines by one to two orders of magnitude. Table 2 shows a comprehensive summary of the performance metrics, with our model achieving an improvement of over 91% across all evaluations for the GS equation.

**FN Equation.** This RD system is another classic but challenging case due to its two-scale fast-slow evolving patterns. As illustrated in the predicted snapshots of Figure 3(**c**), the baseline models can capture the global patterns but struggle with the local details. Our method shows superiority in learning the multi-scale features. Moreover, the error analysis demonstrates that our P$^2$C$^2$Net achieves errors one to two orders of magnitude lower than those of the baseline models, with an improvement ranging from 53.8% to 77.5% compared to the best baseline (see Table 2).

**NS Equation.** We evaluate our method on an NS dataset with a Reynolds number of 1000, a benchmark dataset known for its significant challenges. As shown in Figure 3(**d**), the snapshots produced by the baseline models at $t = 7$ s exhibit incorrect dynamical patterns, especially the LI model showing divergence. The average test error of our model is at least an order of magnitude lower than those of the baselines. Furthermore, $P^2C^2$Net consistently outperforms the baselines throughout the error propagation process. Table 2 further validates the superior performance of our model, with improvements across all metrics of at least 57%. Additionally, we examine the physical properties of the learned fluid dynamics, such as energy spectra. As illustrated in Figure 4, the energy spectra curve of $P^2C^2$Net closely aligns with that of the ground truth, demonstrating its effectiveness in capturing high-frequency features.

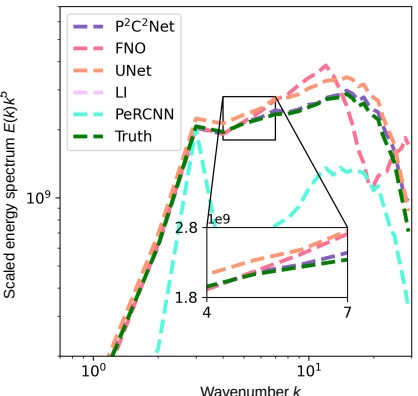

Figure 4: Energy spectra.

**Generalization test.** Taking the NS equation as an example, $P^2C^2$Net is able to generalize to different external forces $\mathbf{f}$ and Reynolds numbers $Re$. Our model is trained with $\mathbf{f} = \sin(4y)\mathbf{n}_x - 0.1\mathbf{u}$ and $Re = 1000$, where $\mathbf{n}_x = [1,0]^T$. We conduct the generalization test under six external force scenarios and four distinct Reynolds numbers $Re = \{200, 500, 800, 2000\}$. As depicted in Figure 5(**a**), the error distribution presents a stable performance with errors generally below 0.1 across six distinct external forces. Figure 5(**b**) shows the acceptable error propagation with a gradual upward trend, which aligns with the long-term rollout outcomes of our model. For the generalization test on Reynolds numbers, Figure 5(**c**) demonstrates satisfactory and robust error levels for all Reynolds numbers, and smaller errors are observed when the Reynolds numbers are closer to those in the training set, as validated by the error propagation curves shown in Figure 5(**d**). Overall, our $P^2C^2$Net model exhibits robust generalization capabilities and stable performance across test samples for external forces, and Reynolds numbers. Further details on the snapshots are provided in Appendix Figure S3.

**Robustness to noisy/incomplete data.** Using the Burgers equation as an example, we introduced Gaussian noise of varying scales during the training process and observed its minimal impact on the results. The results are presented in Table S3, which indicate that our model is robust to the training data noise and maintains the HCT (high correlation time) metric without reduction.

Moreover, for the Burgers case, the time steps of the training data consist of only 28% of the inference steps in the test data. Namely, our training data itself is incomplete, and the missing data accounts for 72% of the test set. Based on this sparse dataset, we further randomly reduced the training data by 20% to simulate data incompleteness (e.g., randomly deleting data according to the rollout step size to make the trajectory incomplete, where the specific size can be found in Table 1 in the paper). The results of this experiment are shown in Table S4, where values are averaged over 10 test sets.

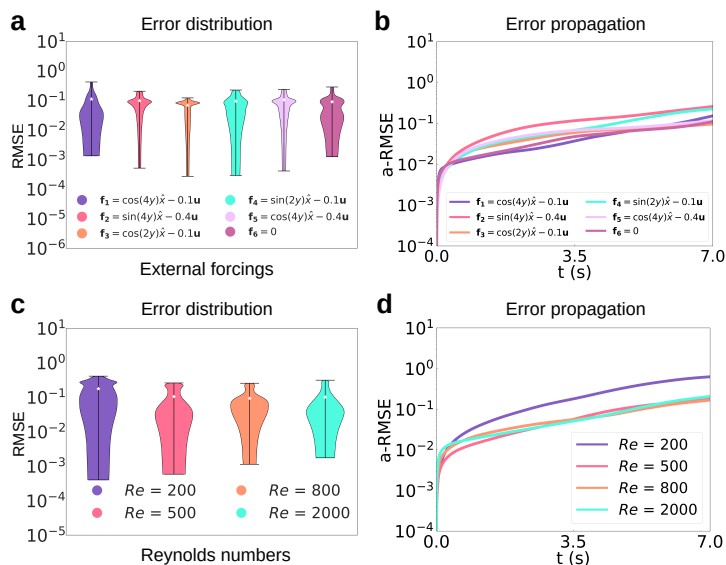

Figure 5: The error distribution and propagation of $P^2C^2$Net for generalization over different external forces (**a**, **b**) and Reynolds numbers (**c**, **d**).

We can observe that, after making the data sparser, our model performance slightly decreases, which means that our model is capable of handling scenarios with incomplete data.

**Ablation study.** To assess the impact of different components on model performance, we design and conduct a series of ablation experiments based on the Burgers equation. The experimental setups include: (1) Model 1, replacing symmetric convolutions with regular convolutions; (2) Model 2, using convolution kernels with finite difference stencils instead of symmetric convolutions; (3) Model 3, removing the Correction Block; (4) Model 4, substituting RK4 integration with first-order Euler methods; and (5) the full $P^2C^2$Net architecture. All experiments are performed using the same training data and model hyperparameters as previously defined.

As shown in Table 3, the network performs poorly when the Fourier block is removed, with error levels two orders of magnitude higher compared to using the complete $P^2C^2$Net architecture. This emphasizes the critical role of the physics-encoded variable correction learning method. Configurations using traditional finite difference stencils as kernels also exhibit simi-

Table 3: Results for the ablation study of $P^2C^2$Net.

| Ablated Model | RMSE | MAE | MNAD | HCT (s) |
|---|---|---|---|---|
| Model 1 | 0.1450 | 0.1100 | 0.0924 | 0.1073 |
| Model 2 | NaN | NaN | NaN | 0.2013 |
| Model 3 | 0.1463 | 0.1141 | 0.0977 | 0.1100 |
| Model 4 | 0.0322 | 0.0241 | 0.0193 | 1.1860 |
| Full $P^2C^2$Net | **0.0064** | **0.0046** | **0.0037** | **1.4000** |

larly poor performance due to the large numerical errors when approximating derivatives caused by coarse grids. Moreover, removing symmetry constraints leads to a significant increase in errors, emphasizing the importance of symmetric features for model convergence, especially in scenarios with limited data. Using the Euler scheme for time stepping instead of RK4 results in reduced stability and increased error accumulation, leading to a decrease in performance compared to the complete $P^2C^2$Net. Overall, the results confirm that the physics-encoded variable correction learning method and convolution filters satisfying symmetry are indispensable components of the network framework.

# 5   Conclusion

This paper presents a physics-encoded variable correction learning method designed to embed prior knowledge on coarse grids for solving nonlinear dynamic systems. This approach enables the model to focus on fitting equations, ensuring excellent generalization capability and interpretability. It introduces a convolutional filter that follows symmetry constraints, requiring only seven learnable parameters to adaptively compute the derivatives of system state vari-

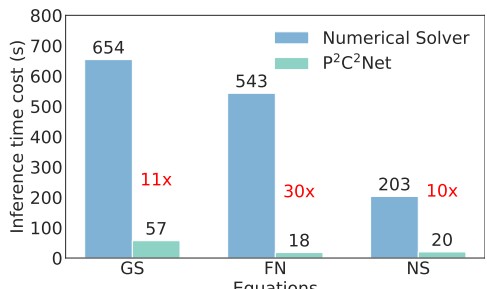

Figure 6: Computational time for comparison.

ables corresponding to the flow field on a coarse grid. This allows the model to learn spatiotemporal dynamics with limited data. Our model has been trained and tested on four different nonlinear dynamic systems, achieving the SOTA results. Even with minimal data, it can generalize to different initial conditions, external force terms, and PDE parameters. In summary, our model effectively adapts to various nonlinear dynamic systems. Moreover, the trained model shows remarkable speedup for simulation under the same condition of accuracy, shown in Figure 6.

Despite demonstrating excellent generalization ability, the model faces two challenges. Firstly, the model is based on regular grids with periodic boundaries, limiting its ability to solve problems on irregular grids. We will explore graph structures or coordinate transformations to handle irregular grids and incorporate special boundary treatment methods to adapt to various boundary conditions. Secondly, we expect to expand our research from 2D problems to 3D dynamical systems.

# Acknowledgement

The work is supported by the National Natural Science Foundation of China (No. 62276269 and No. 92270118), the Beijing Natural Science Foundation (No. 1232009), and the Strategic Priority Research Program of the Chinese Academy of Sciences (No. XDB0620103). In addition, H.S and Y.L. would like to acknowledge the support from the Fundamental Research Funds for the Central Universities (No. 202230265 and No. E2EG2202X2). Q.W. acknowledges the support by the Interdisciplinary-Innovative Research Program of the Institute of Interdisciplinary Sciences, Renmin University of China. P.R. would like to disclose that he was involved in this work when he was at Northeastern University, who has not been supported by Huawei Technologies.

## Impact statement

The aim of this work is to develop a novel physics-encoded learning scheme to accelerate predictions and simulations of spatiotemporal dynamical systems. This method can be applied to various fields, including weather forecasting, turbulent flow prediction, and other simulation tasks. Our research is solely intended for scientific purposes and poses no potential ethical risks.

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

# APPENDIX

## A   Design of the filter with symmetric constraint

Firstly, inspired by the central finite difference method, we design a filter $\mathbf{g}$ with the size of $m \times m$, which meets the symmetric constraint. For example, when $m = 5$, $\mathbf{g}$ is given by

$$\mathbf{g} = \begin{pmatrix} a_1 & a_4 & a_7 & -a_4 & -a_1 \\ a_2 & a_5 & a_8 & -a_5 & -a_2 \\ a_3 & a_6 & 0 & -a_6 & -a_3 \\ -a_2 & -a_5 & -a_8 & a_5 & a_2 \\ -a_1 & -a_4 & -a_7 & a_4 & a_1 \end{pmatrix}. \tag{S1}$$

Taking $\alpha = (1, 0)$, we have:

$$\sum_{k_1=-\frac{m-1}{2}}^{\frac{m-1}{2}} \sum_{k_2=-\frac{m-1}{2}}^{\frac{m-1}{2}} k_1^{\alpha_1} k_2^{\alpha_2} g[k_1, k_2] = -4a_7 + 2a_8. \tag{S2}$$

In order to satisfy the sum rules of order $\alpha = (1, 0)$, we can get $a_8 = -2a_7$. Then, the filter $\mathbf{g}$ can be re-written as

$$\mathbf{g} = \begin{pmatrix} a_1 & a_4 & a_7 & -a_4 & -a_1 \\ a_2 & a_5 & -2a_7 & -a_5 & -a_2 \\ a_3 & a_6 & 0 & -a_6 & -a_3 \\ -a_2 & -a_5 & 2a_7 & a_5 & a_2 \\ -a_1 & -a_4 & -a_7 & a_4 & a_1 \end{pmatrix}. \tag{S3}$$

**Corollary 1 Proof:**   Consider the filter we designed above with $\alpha = (0, 1)$, $\alpha = (0, 3)$ and $\alpha = (3, 0)$, we have:

$$\sum_{k_1=-\frac{m-1}{2}}^{\frac{m-1}{2}} \sum_{k_2=-\frac{m-1}{2}}^{\frac{m-1}{2}} k_1^0 k_2^1 g[k_1, k_2] = -4a_5 - 2a_6 \tag{S4}$$

$$\sum_{k_1=-\frac{m-1}{2}}^{\frac{m-1}{2}} \sum_{k_2=-\frac{m-1}{2}}^{\frac{m-1}{2}} k_1^0 k_2^3 g[k_1, k_2] = -16a_5 - 2a_6 \tag{S5}$$

$$\sum_{k_1=-\frac{m-1}{2}}^{\frac{m-1}{2}} \sum_{k_2=-\frac{m-1}{2}}^{\frac{m-1}{2}} k_1^3 k_2^0 g[k_1, k_2] = -12a_7 \tag{S6}$$

Satisfying the sum rules of order $\alpha = (0, 3)$ and $\alpha = (3, 0)$ leads to strictly $-16a_5 - 2a_6 = 0$, $-12a_7 = 0$. By adjusting the trainable parameters, we have $a_6 + 8a_5 \to 0$, $a_7 \to 0$, and $-4a_5 - 2a_6 \neq 0$. According to Lemma1, the resulting filter $\mathbf{g}$ satisfies the total sum rules of order $5 \setminus \{2\}$.

For any smooth function $\omega$, we perform a convolution operation on it with the filter $\mathbf{g}$. Applying a Taylor expansion to this process, we obtain:

$$\sum_{k_1, k_2=-2}^{2} g[k_1, k_2] \omega(x + k_1 \delta x, y + k_2 \delta y)$$

$$= \sum_{k_1, k_2=-2}^{2} g[k_1, k_2] \sum_{i,j=0}^{4} \frac{\delta x^i \delta y^j}{i! j!} \frac{\partial^{i+j} \omega}{\partial x^i \partial y^j}(x, y) + o(|\delta x|^4 + |\delta y|^4) \tag{S7}$$

$$= \sum_{i,j=0}^{4} r_{i,j} \delta x^i \delta y^j \frac{\partial^{i+j} \omega}{\partial x^i \partial y^j}(x, y) + o(|\delta x|^4 + |\delta y|^4).$$

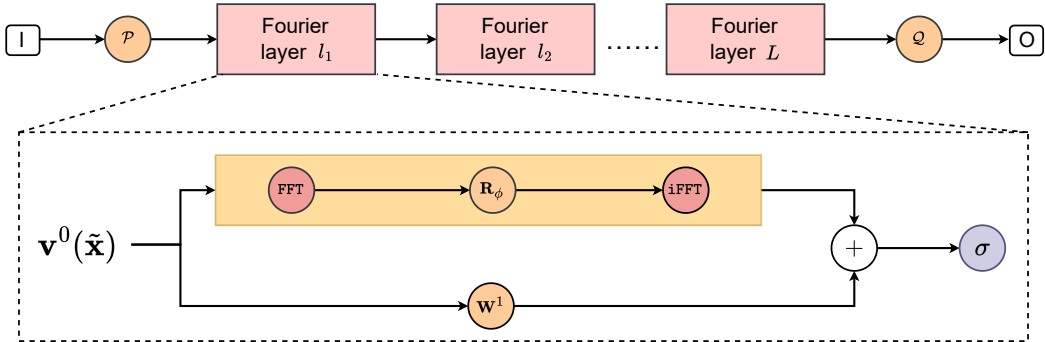

Figure S1: The architecture of FNO Model

Then, if the filter $\mathbf{g}$ has total sum rules of order $5 \setminus \{2\}$, for a smooth function $\omega(x, y)$ and small perturbation $\varepsilon > 0$, we have:

$$\frac{1}{\varepsilon} \sum_{k_1=-\frac{m-1}{2}}^{\frac{m-1}{2}} \sum_{k_2=-\frac{m-1}{2}}^{\frac{m-1}{2}} g[k_1, k_2]\omega(x + \varepsilon k_1, y + \varepsilon k_2) = C\frac{\partial \omega(x, y)}{\partial x} + \mathcal{O}(\varepsilon^4), \text{ as } \varepsilon \to 0. \quad (S8)$$

If there is no such setting, for $a_6 + 8a_5 \to 0$, $a_7 \to 0$, through continuous training, we iteratively optimize the network and adjust the values of parameter $a_1$ to $a_7$, such that:

$$\sum_{i,j=0}^{4} r_{i,j}\delta x^i \delta y^j \frac{\partial^{i+j}\omega}{\partial x^i \partial y^j}(x, y) = 0 \ \forall i, j \setminus \{i = 1, j = 0\} \quad (S9)$$

In this case, Eq. (S8) still holds. Hence, by employing a filter $\mathbf{g}$ with enhanced symmetry constraints and through the learning process of the network, we ensure that fourth-order accuracy is preserved.

## B  Implementation Details

### B.1  FNO Model

As illustrated in Figure S1, the network structure primarily consists of three key components: $\mathcal{P}$ (lift operation), $\mathcal{Q}$ (projection operation), and Fourier layers. Both $\mathcal{P}$ and $\mathcal{Q}$ are convolutional operations designed for channel transformation. Every Fourier layer employs the Fast Fourier Transform (`FFT`) and the inverse Fast Fourier Transform (`iFFT`) for frequency domain transformations. Additionally, $\mathbf{R}_\phi$ represents spectral filtering and convolution in the frequency domain, and $\mathbf{W}^l$ denotes the local linear transformation specific to the $l$-th layer. $\sigma$ the activation function GeLU.

Our FNO model omits normalization schemes due to their detrimental impact on performance [46]. When used in the Correction Block, we set $L = 2$. For the Burgers equation, we configure modes = 12, width = 12, and the projection operation from channel 12 to channel 50. Both the FHN and GS equations share the same configuration: modes = 12, width = 20, with the projection operation from channel 20 to channel 50. The NS equation requires a distinct configuration: modes = 25 width = 20, and the projection operation from channel 20 to channel 128. In particular, when it appears as a module of Neural Network, the hyperparameters are set differently: $L = 5$, modes $= grid/2 - 2$, and width to be the same as modes.

### B.2  Poisson Solver

The objective of the Poisson solver is to determine the state quantities of a system within a two-dimensional spatial domain using the spectral method for a specified Laplacian term. The Laplace equation in two dimensions is represented as:

$$\Delta p = \psi(\mathbf{u}, \mathbf{f}). \quad (S10)$$

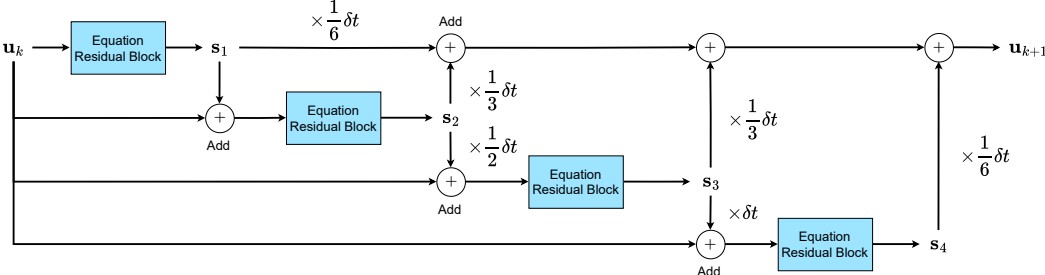

Figure S2: RK4 integration scheme

where $\psi(\mathbf{u}, \mathbf{f}) = -2\left(u_y v_x - u_x v_y\right) + \boldsymbol{\nabla} \cdot \mathbf{f}$ for 2D problems. Applying Fast Fourier Transform (FFT) on Eq. (S10), we have

$$-(\eta_x^2 + \eta_y^2)p^* = \psi^*(\mathbf{u}, \mathbf{f}), \tag{S11}$$

where $\eta_x$ and $\eta_y$ are the wavenumbers along the $x$ and $y$ axes, respectively, assuming $\eta_x^2 + \eta_y^2 \neq 0$ to avoid division by zero. In the frequency domain, we can get

$$p^* = \frac{\psi^*(\mathbf{u}, \mathbf{f})}{-(\eta_x^2 + \eta_y^2)}. \tag{S12}$$

Next, we transform the field from the frequency domain to the spatial using inverse Fast Fourier Transform (iFFT):

$$\texttt{iFFT}\left[p^*\right] = \texttt{iFFT}\left[\frac{\psi^*(\mathbf{u}, \mathbf{f})}{-(\eta_x^2 + \eta_y^2)}\right]. \tag{S13}$$

By applying the above process to $\psi(\mathbf{u}, \mathbf{f})$, we can efficiently decouple the pressure field without any labeled data or training process.

### B.3 RK4 integration scheme

The general numerical integration method for time marching from $\mathbf{u}_{t_k}$ to $\mathbf{u}_{t_{k+1}}$ can be written as:

$$\mathbf{u}_{k+1} = \mathbf{u}_k + \int_{t_k}^{t_{k+1}} \mathcal{H}[\mathbf{u}_k(\mathbf{x}, \tau)]\mathrm{d}\tau. \tag{S14}$$

Among them, $\mathbf{u}_{k+1}$ and $\mathbf{u}_k$ are solutions at time $k + 1$ and $k$. As shown in Figure S2, RK4 is a high-order integration scheme, which divides the time interval into multiple equally spaced small time steps to approximate the integral. The final updating of the above state change can be written as:

$$
\begin{aligned}
\mathbf{s}_1 &= \mathcal{H}\left[\mathbf{u}_k, t_k\right], \\
\mathbf{s}_2 &= \mathcal{H}\left[\mathbf{u}_k + \frac{\delta t}{2} \times \mathbf{s}_1, t_k + \frac{\delta t}{2}\right], \\
\mathbf{s}_3 &= \mathcal{H}\left[\mathbf{u}_k + \frac{\delta t}{2} \times \mathbf{s}_2, t_k + \frac{\delta t}{2}\right], \\
\mathbf{s}_4 &= \mathcal{H}\left[\mathbf{u}_k + \delta t \times \mathbf{s}_3, t_k + \delta t\right], \\
\mathbf{u}_{k+1} &= \mathbf{u}_k + \frac{1}{6}\delta t(\mathbf{s}_1 + 2\mathbf{s}_2 + 2\mathbf{s}_3 + \mathbf{s}_4).
\end{aligned}
\tag{S15}
$$

where $\delta t$ denotes the step size and $\mathbf{s}_1, \mathbf{s}_2, \mathbf{s}_3, \mathbf{s}_4$ represent four intermediate variables (slopes). The global error is proportional to the step size to the fourth power, i.e., $\mathcal{O}(\delta t^4)$.

## C  Baseline models

To evaluate the performance of our proposed method, we compare it against multiple state-of-the-art (SOTA) baseline models. The detailed descriptions of each model are presented below, and the training configurations are listed in Section F.

**Fourier Neural Operator (FNO).** The FNO [8] combines neural networks with Fourier transforms and consists of two major parts. The first part primarily involves performing Fourier transforms on system state quantities, convolving in the frequency domain, and then inversely transforming to extract global features. The second part includes linear transformations through convolutions on the system state quantities to extract local features. Finally, activation functions are applied, and the outputs from both parts are combined to produce the final result.

**UNet.** UNet [47] employs an encoder-decoder structure, where the encoder extracts multi-scale features through multiple downsampling operations and the decoder recovers the original image size through multiple upsampling steps. Skip connections are utilized during the decoding process to fuse feature maps from corresponding layers, preserving both local details and capturing global features.

**PeRCNN.** PeRCNN [19] is a physics-encoded learning approach with physical laws embedded into the neural networks. It comprises multiple parallel CNNs, leveraging the simulation of polynomial equations through feature map multiplication. This approach enhances the model's capacity for extrapolation and generalization.

**DeepONet.** The aim of DeepONet [7] is to use neural networks to learn end-to-end mapping from input data to target output by approximating operators. It includes a trunk net and a branch net, which enables effectively capturing intricate functional relationships.

**Learned Interpolation (LI).** LI [20] is based on a finite volume method scheme. It leverages neural networks to replace the traditional numerical approach that uses polynomial interpolation for the velocity tensor product. This network learns how to dynamically adjust the interpolation function based on the characteristics of the local flow field. Therefore, the LI method facilitates the predictions of dynamics on coarser grids.

# D  Evaluation Metrics

In this paper, we use several metrics to evaluate our model, including RMSE, Mean Absolute Error (MAE), Mean Normalized Absolute Difference (MNAD), and High Correction Time (HCT). RMSE measures the average magnitude of the error between predicted and observed values, reflecting the model's accuracy. MAE evaluates the average absolute difference between predicted and observed values, indicating the actual magnitude of the errors. MNAD is a crucial metric for assessing the accuracy of model outputs over time. MNAD calculates the average discrepancy across a series of temporal data points, offering a normalized measure of prediction error relative to the range of the actual data. HCT quantifies the model's ability to make accurate long-term predictions. These metrics are defined as follows:

$$\text{RMSE} = \sqrt{\frac{1}{n}\sum_{i=1}^{n}\|\mathbf{S}_i - \check{\mathbf{S}}_i\|^2}, \tag{S16}$$

$$\text{MAE} = \frac{1}{n}\sum_{i=1}^{n}\left|\mathbf{S}_i - \check{\mathbf{S}}_i\right| \tag{S17}$$

$$\text{MNAD} = \frac{1}{n}\sum_{i=1}^{n}\frac{\|\mathbf{S}_i - \check{\mathbf{S}}_i\|}{\|\mathbf{S}_i\|_{\max} - \|\mathbf{S}_i\|_{\min}}, \tag{S18}$$

$$\text{HCT} = \sum_{i=1}^{N}\Delta t \cdot \mathbf{1}(PCC(\mathbf{S}_i, \check{\mathbf{S}}_i) > 0.8), \tag{S19}$$

where

$$PCC(\mathbf{S}_i, \check{\mathbf{S}}_i) = \frac{\text{cov}(\mathbf{S}_i, \check{\mathbf{S}}_i)}{\sigma_{\mathbf{S}_i}\sigma_{\check{\mathbf{S}}_i}}, \tag{S20}$$

Here $n$ denotes the number of trajectories; $\mathbf{S}_i$ the ground truth of trajectories; $\hat{\mathbf{S}}_i$ the spatiotemporal sequence predicted by the model. "cov" is the covariance function and "$\sigma$" is the standard deviation of the given sequence. $\mathbf{1}(\cdot)$ is the indicator function that takes a value of 1 when the condition $(PCC(\mathbf{S}_i, \check{\mathbf{S}}_i) > 0.8)$ is true, and 0 otherwise. $N$ is the total number of time steps.

# E   Dataset Informations

To facilitate a comprehensive evaluation of our proposed method, we employ datasets generated from various well-known equations that model complex systems. These datasets cover a wide range of applications from fluid dynamics to reaction-diffusion systems, ensuring a robust assessment of model performance.

For fairness of testing datasets, we randomly select 10 seeds from the range of [1, 10,000], which are used to generate different Gaussian noise disturbances, which are then applied to the initial velocity field to create varying ICs. We also perform a warm-up phase during data generation to ensure that the variance and mean of the trajectories are closely aligned, thereby maintaining fairness. More details about dataset generation can be found in Table S2.

**Burgers.** This equation describes the velocity flow in a viscous fluid, which combines compressibility and nonlinear effects. It has wide applications in many scientific fields, including weather and climate simulation, the petroleum industry, acoustics, and quantum field theory. The equation is given by:

$$\frac{\partial \mathbf{u}}{\partial t} = \nu \boldsymbol{\nabla}^2 \mathbf{u} - \mathbf{u} \cdot \boldsymbol{\nabla} \mathbf{u}, \quad t \in [0, T] \tag{S21}$$

where $\mathbf{u} = \{u, v\} \in \mathbb{R}^2$ represents the fluid velocities, $\nu$ is the viscosity coefficient set to 0.002, and $\boldsymbol{\nabla}^2$ is the Laplacian operator.

To generate the dataset, we use the FD method with periodic boundary conditions on the spatial domain $\mathbf{x} \in [0, 1]$. The data is initially simulated on a $100^2$ grid and then downsampled to $25^2$ for numerical experiments. The simulation timestep is set to $1 \times 10^{-3}$ seconds and the time duration is $T = 1.4$ s. We use five trajectories for training, each of which contains 400 snapshots. For testing, we utilize ten groups of trajectories, each with 1400 snapshots.

**GS.** The GS equation models the interaction between two chemical species and has been widely used to simulate various chemical reactions under different conditions. It helps to explore dynamic phenomena such as self-organized patterns, chemical waves, reaction-diffusion patterns, collective behavior, natural pattern formation, cell growth, and other areas in physics. The GS equation is described as:

$$\begin{aligned}
\frac{\partial u}{\partial t} &= D_u \boldsymbol{\nabla}^2 u - uv^2 + F(1 - u), \\
\frac{\partial v}{\partial t} &= D_v \boldsymbol{\nabla}^2 v + uv^2 - (F + \kappa)v,
\end{aligned} \tag{S22}$$

where $u$ and $v$ represent the concentrations of two different chemical substances, $D_u$ and $D_v$ are the diffusion coefficients, and $F$ and $\kappa$ denote the growth and death rates of the substances, respectively. In our case, we set $D_u = 2.0 \times 10^{-5}$, $D_v = 5.0 \times 10^{-6}$, $F = 0.04$, and $\kappa = 0.06$. We create the dataset using the FD method on a $128^2$ grid under periodic boundary conditions, with the spatial domain $\mathbf{x} \in [0, 1]^2$. The simulation timestep is 0.5 s, and the total simulation duration is 1900 s. The data is then downsampled to a $32^2$ grid and the timestep is coarsened to 2 s to establish the ground truth. We leverage three training trajectories, each containing 1000 snapshots. Additionally, we employ ten separate testing sets with diffenent ICs.

**FN.** This case is a widely recognized reaction-diffusion system and is used to simulate the propagation of neural impulses. The governing equations are given by:

$$\frac{\partial \mathbf{u}}{\partial t} = \gamma \nabla^2 \mathbf{u} + \mathbf{M}(\mathbf{u}), \quad t \in [0, T] \tag{S23}$$

where $\mathbf{u} = \{u, v\} \in \mathbb{R}^2$ are the two interactive components, and $\gamma$ is the diffusion coefficient. The reaction source terms $\mathbf{M}(\mathbf{u})$ are defined as:

$$\begin{aligned}
M_u(u, v) &= u - u^3 - v + \alpha, \\
M_v(u, v) &= \beta(u - v),
\end{aligned} \tag{S24}$$

with $\alpha = 0.01$ denoting the external stimulus and $\beta = 0.25$ as the reaction coefficient.

We generated the dataset using the FD method on a $128^2$ grid and a timestep of $2.0 \times 10^{-3}$ s. The spatial domain was set as $\mathbf{x} \in [0, 128]^2$ with periodic boundary conditions applied. The simulation

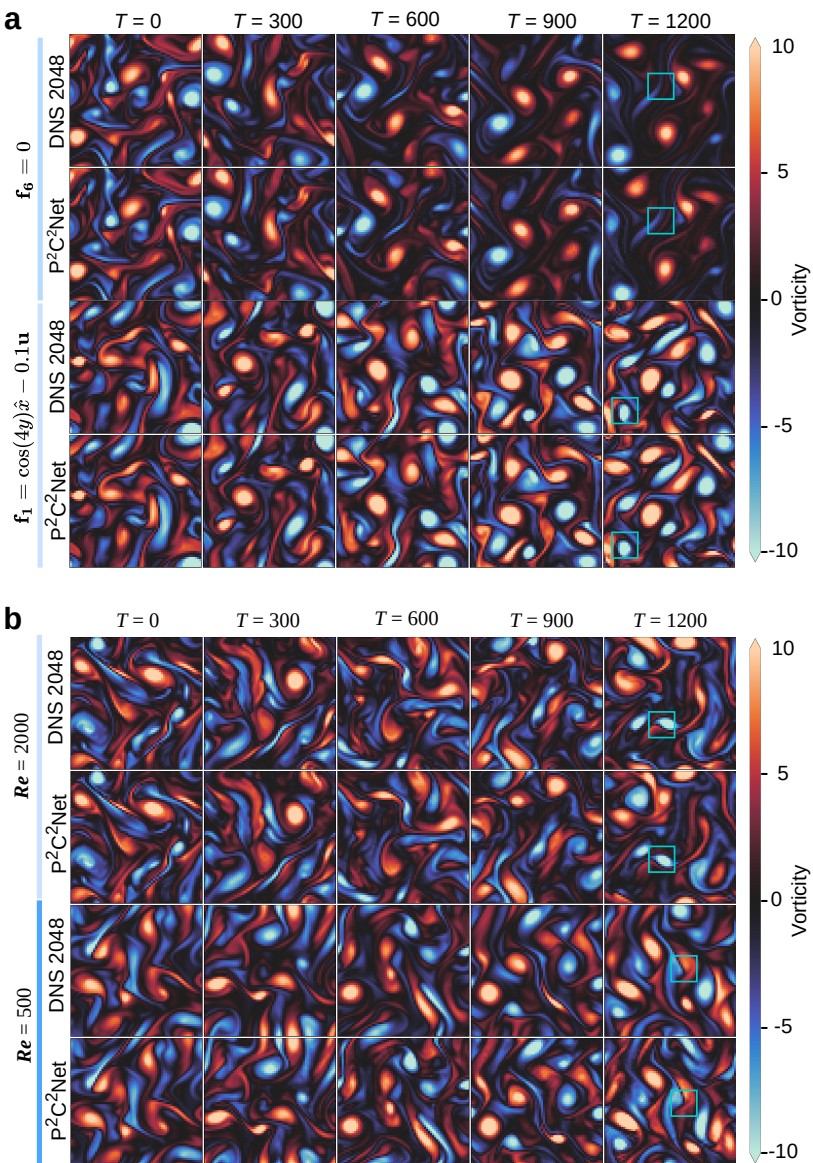

Figure S3: Generalization across multiple $Re$'s and forces.

time is $T = 10$ s. This data is subsequently downsampled to $64^2$ with a timestep of $8.0 \times 10^{-3}$ s to serve as the ground truth. For training purposes, we employ three groups of trajectories, each comprising 1375 snapshots. For our testing phase, we leverage ten distinct groups of trajectories, each with unique ICs.

**NS.** The NS equations are fundamental to the study of fluid dynamics, which govern the behavior of fluid motion. In this paper, we consider a 2-dimensional incompressible Kolmogorov flow with periodic boundary conditions in velocity-pressure form, which is written as

$$\frac{\partial \mathbf{u}}{\partial t} + (\mathbf{u} \cdot \boldsymbol{\nabla})\mathbf{u} = \frac{1}{Re}\boldsymbol{\nabla}^2 \mathbf{u} - \boldsymbol{\nabla}p + \mathbf{f}, \quad t \in [0, T],$$
$$\boldsymbol{\nabla} \cdot \mathbf{u} = 0, \tag{S25}$$

where $\mathbf{u} = \{u, v\} \in \mathbb{R}^2$ denotes the fluid velocity vector, $p \in \mathbb{R}$ is the pressure, $Re$ representes the Reynolds number that characterizes the flow regime. The Reynolds number serves as a scaling parameter in the NS equation, resulting in a balance between the inertial forces (captured by the advection term $(\mathbf{u} \cdot \boldsymbol{\nabla})\mathbf{u}$) and the viscous forces(captured by the Laplacian term $\boldsymbol{\nabla}^2 \mathbf{u}$). Thus, for

Table S1: Impact of different kernel sizes.

| Filter | RMSE | MAE | MNAD | HCT |
|--------|------|-----|------|-----|
| $3 \times 3$ | 0.1274 | 0.0926 | 0.0773 | 0.1947s |
| $5 \times 5$ | 0.0064 | 0.0046 | 0.0037 | 1.4s |
| $7 \times 7$ | NaN | NaN | NaN | 0.1673s |

Table S2: Dataset generate details

| Parameters / Case | Burgers | GS | FN | NS |
|-------------------|---------|-----|-----|-----|
| DNS Method | FD | FD | FD | FV |
| Spatial Domain | $[0, 1]^2$ | $[0, 1]^2$ | $[0, 128]^2$ | $[0, 2\pi]^2$ |
| Calculate Grid | $100^2$ | $128^2$ | $128^2$ | $2048^2$ |
| Training Grid | $25^2$ | $32^2$ | $64^2$ | $64^2$ |
| Simulation $\delta t$ (s) | $1.00 \times 10^{-3}$ | $2.00 \times 10^{-3}$ | $5.00 \times 10^{-1}$ | $2.19 \times 10^{-4}$ |
| Warmup (s) | 0.1 | 0 | 9 | 40 |
| Training data group | 5 | 3 | 3 | 5 |
| Testing data group | 10 | 10 | 10 | 10 |
| Spatial downsample | $16\times$ | $16\times$ | $4\times$ | $1024\times$ |
| Temporal downsample | $1\times$ | $4\times$ | $4\times$ | $32\times$ |

low Reynolds numbers (i.e., when $Re$ is small), the viscous forces are dominant, and the fluid flow is mostly laminar and smooth. On the other hand, at high Reynolds numbers (i.e., when $Re$ is large), the inertial forces are dominant, and the fluid flow becomes more turbulent and chaotic.

In this paper, we generate the training data for a $Re$ = 1000, using periodic boundary conditions within the spatial domain $\mathbf{x} \in [0, 2\pi]^2$. The data generation employs the FV method on a $2048^2$ grid with a simulation timestep of $2.19 \times 10^{-4}$ s. Subsequently, this data is downsampled to $64^2$ with a time step of $7.0 \times 10^{-3}$ to produce the ground truth. Five groups of trajectories, each containing 4800 snapshots, are employed for training. Ten groups of trajectories with diverse ICs are used for each of the tests. Specifically, to further test the model's generalization ability, we expand the test set: (1) generating data corresponding to $Re$ values of 200, 500, 800, 1000, and 2000, and (2) creating data with different external forces:

$$\begin{aligned}
\mathbf{f}_1 &= \cos(4y)\mathbf{n}_x - 0.1\mathbf{u}, \\
\mathbf{f}_2 &= \sin(4y)\mathbf{n}_x - 0.4\mathbf{u}, \\
\mathbf{f}_3 &= \cos(2y)\mathbf{n}_x - 0.1\mathbf{u}, \\
\mathbf{f}_4 &= \sin(2y)\mathbf{n}_x - 0.1\mathbf{u}, \\
\mathbf{f}_5 &= \cos(4y)\mathbf{n}_x - 0.4\mathbf{u}, \\
\mathbf{f}_6 &= 0
\end{aligned} \tag{S26}$$

## F  Training Details

All experiments were conducted on a single 80GB Nvidia A100 GPU, using an Intel(R) Xeon(R) Platinum 8380 CPU (2.30GHz, 64 cores). We only give some of the changed parameters here, and the other hyperparameters remain the same as the default settings.

**P$^2$C$^2$Net.**  The architecture of P$^2$C$^2$Net, as shown in Figure 1, utilizes the Adam optimizer with a learning rate set at $5 \times 10^{-3}$. The model is trained with a batch size of 16 over 500 epochs. The rollout timestep settings are detailed in Table 1. We employ the StepLR scheduler scaling the learning rate by 0.96 every 200 steps.

**FNO.**  The network architecture of FNO remains largely consistent with the original paper [8], with the primary modification being the adaptation of the training method for this model to an autoregressive approach. We utilize the Adam optimizer, with a learning rate of $1 \times 10^{-3}$ and a

Table S3: Impact of noise on P$^2$C$^2$Net performance

| Training | RMSE | MAE | MNAD | HCT (s) |
|---|---|---|---|---|
| + 1% noise | 0.0092 | 0.0088 | 0.0062 | 1.4 |
| + 0.5% noise | 0.0078 | 0.0057 | 0.0047 | 1.4 |
| w/o Noise noise | 0.0064 | 0.0046 | 0.0037 | 1.4 |

Table S4: Impact of sparser Burgers dataset on P$^2$C$^2$Net performance

| Training | RMSE | MAE | MNAD | HCT (s) |
|---|---|---|---|---|
| reduce 20% | 0.0073 | 0.0052 | 0.0050 | 1.4 |
| 5×400 snapshots (in paper) | 0.0064 | 0.0046 | 0.0037 | 1.4 |

batch size of 20. The training spans 1000 epochs, and the rollout timestep is aligned with that of the P$^2$C$^2$Net.

**UNet.** We employ the modern UNet [22] architecture with the default setting, with the rollout timestep consistent with that of P$^2$C$^2$Net. The scheduler utilized is StepLR with a step size of 100 and a gamma of 0.96. The optimizer employed is Adam, with a learning rate of $1 \times 10^{-3}$ and a batch size of 10. The training is conducted over 1000 epochs.

**DeepONet.** We adopt the default architecture of DeepONet [7], with the Adam optimizer. The learning rate is set at $5 \times 10^{-4}$, with a decay applied every 5000 steps scaling at 0.9. The training is conducted with a batch size of 16 and spans 20000 epochs.

**PeRCNN.** We utilize an architecture that is identical to the standard PeRCNN configuration [19]. The optimization is performed using Adam, with a StepLR scheduler that reduces the learning rate by a factor of 0.96 every 100 steps. The initial learning rate is set at 0.01. The training regimen spans 1000 epochs with a batch size of 32.

**LI.** We utilizes its default network architecture and parameter configurations [20]. The optimizer chosen is Adam with $\beta_1 = 0.9$ and $\beta_2 = 0.99$. The batch size is set to 8, with a global gradient norm clipping of 0.01. The learning rate is $1 \times 10^{-3}$, and the weight decay is $1 \times 10^{-6}$.

# G  Additional Results

## G.1  Applicability to different BCs

To verify the applicability of our model to different BCs, we use the Burgers equation as an example and set the left boundary as Dirichlet, the right boundary as Neumann, and the top/bottom boundaries as Periodic. Here, we denote this case as Complex Boundary Conditions (CBC). The rest of the data generation setup (e.g., ICs, mesh grids) remains the same as used in the paper. We generated 10 CBC test datasets resulting from different random ICs.

We then directly tested the model previously trained based on Periodic BC datasets reported in the paper, meanwhile processing the boundaries using the BC encoding strategy [19] during inference. The quantitative results (average over 10 datasets) are presented in Table S5 where we also list the predicted snapshots at 1.4 s for two random ICs in Figure S4 . We can see that our model is capable of generalizing over different BCs.

## G.2  Generalization Results

For the NS equation, we also evaluate the generalization capability of P$^2$C$^2$Net with respect to multiple $Re$'s and forces. Please refer to Figure S3 for details.

Table S5: Generalization of $P^2C^2$Net over different boundaries on the Burgers example for 10 trajectories.

| BC Type | RMSE | MAE | MNAD | HCT(s) |
|---------|------|-----|------|--------|
| Complex | 0.0202 | 0.0103 | 0.0087 | 1.4 |
| Periodic | 0.0064 | 0.0046 | 0.0037 | 1.4 |

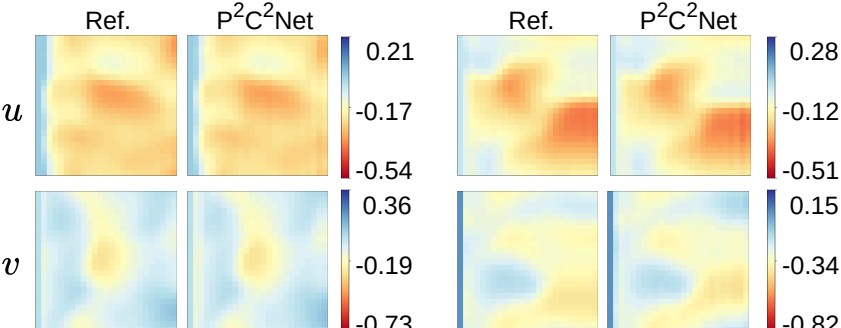

Figure S4: $P^2C^2$Net generalization over complex BCs (left Dirichlet, right Neumann, top/bottom Periodic) on the Burgers example for two random ICs. Snapshots at $t = 1.4$ s.

