# OpenReview forum: "P$^2$C$^2$Net: PDE-Preserved Coarse Correction Network for efficient prediction of spatiotemporal dynamics"
_NeurIPS.cc/2024/Conference — NeurIPS 2024 poster_

### Official Review · Reviewer_bvnV · 2024-06-22

**Soundness:** 2
**Presentation:** 3
**Contribution:** 2
**Rating:** 6
**Confidence:** 5

**Summary:**

The authors present a P^2C^2Net framework for efficiently solving and simulating PDEs, specifically on coarse grids. To overcome the challenges of simulating PDEs on coarse grids, including the difficulty of estimating numeric derivatives and the inaccuracies in the right-hand side of PDEs on coarse grids, the P^2C^2Net architecture comprises the following modules: (1) an RK4 integration scheme to advance in time, (2) a learnable PDE block that approximates the true dynamics on coarse grids, (3) a symmetric convolutional filter that learns numerical derivatives from data, and (4) a neural corrector module. When modeling the Navier-Stokes equations, an optional Poisson block can be included. The proposed network has shown promising results on several classical benchmark problems, including the Burgers' equation and Navier-Stokes equations, compared to standard baselines in the literature.

**Strengths:**

The authors present the P^2C^2Net framework for the simulation of PDEs on coarse grids, which is data-efficient. Several modules are incorporated, including the learnable symmetric convolutional filters and a neural correction module to overcome the insufficient information from the low-resolution data. Extensive experimental examples performed by the authors have shown that the architecture demonstrates better accuracy than existing baselines, including Fourier Neural Networks and PeRCNN. In addition, the model achieves satisfactory performance on out-of-distribution tests, i.e., the network shows promising results with different Reynolds numbers in the Navier-Stokes equation example.

**Weaknesses:**

Although the structure of the network is clearly presented in the paper, some issues remain to be addressed. First, the Poisson solver and neural correction module are both Fourier Neural Networks. This raises the concern that the advantage of the model may stem from its larger size compared to other baseline models. Although an ablation study is performed on the importance of different modules, it lacks a direct comparison with other models regarding the sizes of the model's parameters. Secondly, the training efficiency is questionable. From the paper, it appears that the proposed model is much larger than the considered baselines. Overall, the efficiency of the proposed model remains unclear, making it difficult to conclude whether the architecture design or simply the size of the model is responsible for its superior performance compared to other baselines.

**Questions:**

1. What is the training time for the proposed network on all benchmark examples? How efficient is it compared to other baselines?
2. How robust is the model to noisy or incomplete training data?
3. How does the proposed network handle different boundary conditions beyond the periodic ones discussed in the paper?

**Limitations:**

Yes, the authors did address the limitations.

---

> ### Author Rebuttal · Authors · 2024-08-07
>
> Thank you for achknowledging the data-efficiency and satisfactory performance on OOD tests of our work. We address your concerns as follows.
>
> ### Weakness
> > **Q1(a). Concerns on the larger size of the Poisson solver and neural correction modules compared to baselines.**
>
> The Poisson solver is essentially a numerical method (e.g., a spectral solver). The basic idea is to convert the original problem into the frequency domain, where the spatial differential operation becomes a multiplication operation via Fourier transform. The Poisson equation is then solved in the frequency domain, and the inverse Fourier transform is then used to obtain the solution in the original spatial domain. This entire process does not involve a Fourier neural network or any learnable parameters.
>
> The correction block utilizes only two Fourier layers that retain a few number of modes, and is therefore not considered a large-scale neural network. Given the unique network architecture, our model achieves good generalization performance under a small and sparse data environment, highlighting its advantages over existing baseline models. For your reference, we show the number of model parameters in **Table R1** in the *1-page PDF rebuttal file*.
>
> > **Q1(b). Training efficiency of the proposed method.**
>
> Please see **our response to Q4 in General Reply Q4** for more details.
>
> ### Questions
> > **Q1. Training time on all benchmark examples compared to baselines.**
>
> Excellent comment! The training time for different mothods is shown in **Table A** below. It can be observed that the training time of our model is generally acceptable. Note that the training has been performed on a single 80GB Nvidia A100 GPU, as described in Appendix F in the paper.
>
> **Table A.** Training time for different models.
>
> | Training cost | Burgers | FN | GS | NS |
> | -------- | -------- | -------- | -------- |-------- |
> |   P$^2$C$^2$Net (ours)  | 2h     |  0.6h   |  2.5h    | 69h  |
> |  FNO   | 1.5h     |  2.2h    |  2.1h    | 33h  |
> |  UNet   | 8.7h     |  15.3h    |  14.1h    | 178h  |
> |  DeepONet   | 0.7h     |  0.9h    |  1.0h    | -  |
> |  PeRCNN   | 6.1h    | 12.4 h    |  10.5h    | 73h  |
> |  LI   | -     |  -    |  -    | 67h  |
>
> > **Q2. Robustness to noisy or incomplete training data.**
>
> Great remark! Using the Burgers equation as an example, we introduced Gaussian noise of varying scales during the training process and observed its minimal impact on the results. The results are presented in **Table B** below, which indicate that our model is robust to certain noise and maintains the HCT (high correlation time) metric without reduction.
>
> **Table B.** Impact of noise on P$^2$C$^2$Net performance.
> | Training | RMSE | MAE | MNAD | HCT (s)|
> | -------- | -------- | -------- | -------- |-------- |
> | + 1%   noise     | 0.0092     | 0.0088     | 0.0062     |1.4     |
> | + 0.5% noise    | 0.0078     | 0.0057     | 0.0047     |1.4     |
> | + 0.3% noise    | 0.0072     | 0.0050     | 0.0041     |1.4     |
> | w/o Noise     | 0.0064     | 0.0046     | 0.0037     |1.4
>
> Moreover, for the Burgers case, the time steps of the training data consist of only 28% of the inference steps in the test data. Namely, our training data itself is incomplete, and the missing data accounts for 72% of the test set. Based on this sparse dataset, we further randomly reduced the training data by 20% to simulate data incompleteness (e.g., randomly deleting data according to the rollout step size to make the trajectory incomplete, where the specific size can be found in Table 1 in the paper). The results of this experiment are shown in **Table C** below, where values are averaged over 10 test sets. Additionally, we conducted the same experiment on the GS dataset, with the results presented in **Table D** below. We can observe that, after making the data sparser, our model performance slightly decreases, which means that our model is capable of handling scenarios with incomplete data.
>
> **Table C.** Impact of sparser Burgers dataset on P$^2$C$^2$Net performance.
>
> | Training data | RMSE | MAE | MNAD | HCT (s)|
> | -------- | -------- | -------- | -------- |-------- |
> | reduce 20%   |  0.0073    | 0.0052    |  0.0050   |1.4     |
> |5x400 snapshots(in paper)| 0.0064  | 0.0046  | 0.0037  | 1.4 |
>
> **Table D.** Impact of sparser GS dataset on P$^2$C$^2$Net performance.
>
> | Training data | RMSE | MAE | MNAD | HCT (s) |
> | -------- | -------- | -------- | -------- |-------- |
> | reduce 20%   |  0.0182    |   0.0091  |  0.0093   | 2000    |
> | 3x1000 snapshots(in paper)   | 0.0135     |  0.0062   |  0.0062   | 2000   |
>
> > **Q3. Applicability to different BCs beyond the periodic BCs.**
>
> Despite we showcase the eficacy of our model on datasets with Periodic BCs, it is applicable to handle other types of BCs, such as Dirichlet and Neumann BCs (see **Table R2**, **Table R5** and **Fig. R1** in the *1-page PDF rebuttal file*). **Please see our detailed response to Q1 in General Reply**.
>
> **Remark:** We sincerely thank you for yopur constructive comments. We hope the above responses are helpful to clarify your questions. We will be happy to hear your feedback and look forward to addressing any additional questions. Your consideration of improving the rating of our paper will be much appreciated!

---

> ### Author Response · Authors · 2024-08-10
> **Sincerely looking forward to your feedback**
>
> Dear Reviewer bvnV,
>
> We hope our point-by-point response in the rebuttal has addressed your concerns. We are very much looking forward to your feedback during the discussion period. We would be more than happy to answer any further questions you may have.
>
> Best regards,
>
> The Authors

---

> ### Author Response · Authors · 2024-08-11
> **Look forward to further discussions before the end of the discussion period**
>
> Dear Reviewer bvnV:
>
> As the author-reviewer discussion period will end soon, we would appreciate it if you could kindly review our responses at your earliest convenience. If there are any further questions or comments, we will do our best to address them before the discussion period ends.
>
> Thank you very much for your time and efforts!
>
> Sincerely,
>
> The Authors

---

### Official Review · Reviewer_aBLf · 2024-07-02

**Soundness:** 3
**Presentation:** 3
**Contribution:** 2
**Rating:** 6
**Confidence:** 4

**Summary:**

The paper solves the problem of predicting complex spatiotemporal dynamics on coarse mesh grids with only a small set of data. It proposed a learnable symmetric Conv filter to estimate the spatial derivatives on coarse grids and incorporated RK4 for correcting coarse solution at low resolution.

**Strengths:**

The problem is well-formulated with real applications. A innovative structured Conv filter is proposed to estimate the spatial derivatives. The effectiveness of the method is demonstrated through extensive and robust experimental results.

**Weaknesses:**

1, The proposed scheme has a lot of components but the connection among these blocks are not obvious and the necessities of having all these components is not really justified. For example, what is the difference between correction block and the NN block? How is this NN block learned on the fly exactly?

2, Could you please provide some details about the difference between the proposed method and PERCNN except for the poisson block?

3, In comparison with other baseline models, what adaptation is used on example NS? For the proposed model, NS requires an additional NN block.
I’m not convinced that without specific adaptation of other models to this example is a fair comparison.

**Questions:**

In addition to the Weakness, I have the following two questions regarding the implementation:

1, To train all the components, is it an end-to-end training or each block is trained individually?

2,  For the generalization test, did more training was done to test on six external force scenarios and Reynolds numbers?

**Limitations:**

The authors adequately addressed the limitations.

---

> ### Author Rebuttal · Authors · 2024-08-07
>
> Thank you for acknowledging the novelty and the effectiveness of our proposed method. We address your concerns as follows.
>
> ### **Weakness**
> > **Q1. Clarifications of each block component, their connections and differences.**
>
> The clarifications of each block component and the connections can be found in **our response to Q2 in General Reply**.
>
> It is noted that the correction block aims to address the information loss that occurs due to resolution reduction before calculating derivatives, helping the model adapt to the coarse grid. During training, it serves as a scaling factor for the derivative term. Achieving this with only a shallow network (2 layers with modes 12) is feasible. However, since the network is trained on a coarse grid with a high spatial coarsening ratio (up to $32^2=1024$ on NS), the reduced high-resolution values do not fully satisfy the governing equation. To mitigate error accumulation during the rollout prediction, compensating the PDE block is necessary. This requires a deeper network (e.g., 4 to 6 layers of FNO).
>
> > **Q2. Difference between our method and PeRCNN except for the Poisson block.**
>
> The differences between these two methods are listed as follows.
>
> -PeRCNN utilizes feature map multiplication to build polynomial combinations (with redundant terms) to approximate the underlying governing equation, which has great performance given *fine mesh grids*. In contrast, our proposed model directly incorporates the governing PDE, preserving its complete and precise form, to predict spatiotemporal dynamics on *coarse mesh grids* under a *small and sparse data environemnt*.
>
> -Learning the diffusion term (e.g., $\Delta \mathbf{u}$) in the governing PDE is challenging. PeRCNN employs fixed finite difference Conv to compute the diffusion term, leading to significant errors. Our approach introduces a learnable Conv with symmetric constraint to compute the diffusion term, effectively mitigating errors on the coarse grid.
>
> -PeRCNN is limited to testing simple cases (e.g., small domains and no external forces) and struggles with generalization to external forces and Reynolds number. Finally, PeRCNN aims to learn high-resolution dynamics with an initial state generator super-resolving from low-resolution grids.
>
> Hence, our model is fundamentally different from PeRCNN.
>
> > **Q3. Adaptations for NS equations and fair comparisons by adding NN blocks for baselines.**
>
> The NN block is a flexible component of our model that can be included or excluded during training based on demand. Since the PDE block alone is sufficient to achieve the state-of-the-art performance on Burgers, FN, and GS datasets, we chose to remove the NN block in these cases to establish a lighter model. However, for the NS equation, given the complexity of the flow ($Re=1000$), we need the NN block as a supplement to alleviate the error accumulation of long-term prediction and the instability issue on coarse grids.
>
> To address the fairness comparison issue, we integrated the NNblock into both UNet and FNO models. The results, presented in the **Table R1** in the *1-page PDF rebuttal file*, show that while NNblock enhances the performance of both models, the improvements are still significantly lower compared to our proposed model.
>
> ### **Questions**
> > **Q1. End-to-end training or individual training?**
>
> Good question! Our model features end-to-end training, making the training process user-friendly and avoiding separate training of individual modules. This also can lead to better overall performance and coherence in the model's outputs, as the interdependencies between different parts of the model are explicitly accounted for during training. Additionally, the inclusion of prior knowledge facilitates the model convergence as shown in **Fig. R4** in the *1-page PDF rebuttal file*. We will clarify this in our revised paper.
>
> > **Q2. More training for generalization tests, such as external force scenarios and Reynolds numbers.**
>
> No additional training was performed for the generalization tests. This aspect has been detailed in our paper, specifically in Section 4.2 (Generalization Test). The model was trained solely on five training sets (trajectories) with Reynolds number $Re=1000$ and external force $\mathbf{f} = \mathrm{sin}(4y)\mathbf{n}_{\mathit{x}} - 0.1\mathbf{u}$. Namely, once our model is trained, it can generalize to the different ICs, Reynolds numbers, and external force terms as shown in the paper.
>
>
> **Remark:** We sincerely thank you for putting forward constructive comments/suggestions. We hope the above responses are helpful to clarify your questions. We will be happy to hear your feedback and look forward to addressing any additional questions. Your consideration of improving the rating of our paper will be much appreciated!

---

> ### Author Response · Authors · 2024-08-10
> **Sincerely looking forward to your feedback**
>
> Dear Reviewer aBLf,
>
> We hope our point-by-point response in the rebuttal has addressed your concerns. We are very much looking forward to your feedback during the discussion period. We would be more than happy to answer any further questions you may have.
>
> Best regards,
>
> The Authors

---

> ### Author Response · Authors · 2024-08-11
> **Look forward to further discussions before the end of the discussion period**
>
> Dear Reviewer aBLf:
>
> As the author-reviewer discussion period will end soon, we would appreciate it if you could kindly review our responses at your earliest convenience. If there are any further questions or comments, we will do our best to address them before the discussion period ends.
>
> Thank you very much for your time and efforts!
>
> Sincerely,
>
> The Authors

---

> > ### Comment · Reviewer_aBLf · 2024-08-12
> >
> > Many thanks for the detailed response. My concerns and questions are addressed and I find the contributions sufficient to raise my score.

---

> > > ### Author Response · Authors · 2024-08-12
> > > **Thank you for increasing the score!**
> > >
> > > Thank you for your positive feedback and for increasing the score. We will include the additional experiments and text in the revised paper.

---

### Official Review · Reviewer_WfL9 · 2024-07-09

**Soundness:** 2
**Presentation:** 3
**Contribution:** 2
**Rating:** 6
**Confidence:** 4

**Summary:**

The paper introduces the $P^2C^2Net$, which is designed to solve spatiotemporal partial differential equations (PDEs) using minimal training data. The architecture consists of two main components: a trainable PDE block and a neural network block. The trainable PDE block updates the coarse solution using a high-order numerical scheme with boundary condition encoding. The neural network block corrects the solution.

**Strengths:**

1. The model integrates physics knowledge directly into the network architecture, improving interpretability and generalizability, especially with limited data.

2. The model achieves consistent state-of-the-art performance with over 50% gain.

**Weaknesses:**

1. The data generation process is not well explained. For instance, the training dataset for the Gray Scott model only includes three trajectories. What are the initial conditions for these three training samples and the ten testing samples? Initial conditions can lead to completely different patterns for the steady states. It is hard to believe that the model can generalize well with only three training samples unless it heavily relies on the classical solver. Even in this case, the improvement based on classical solver should relies on extensive data for generalization.

2. What is the difference between $\tilde{x}$ and $x$. The definitions are not clear. Additionally, why do you use $u$ in equation (1) and use $\hat{u}$ in equaition (3). Even though some of the functionals are learnable, rigorouse definitions should be given.

3. The comparison between P2C2Net and models like FNO and UNet is not fair because those models are purely data-driven, while P2C2Net incorporates classical solvers. There should be a performance comparison and discussion between P2C2Net and classical solvers. Additionally, the current runtime comparison in Figure 6 lacks detailed settings and hardware configuration.

**Questions:**

1. What is the purpose of the lower path in the architecture of model shown in Figure 1(a)? The architecture already includes Poisson and correction blocks in the upper path. More explanation is needed to understand the function and necessity of this lower path.

2. How are the convolution filters and the correction block initialized? This information is crucial for understanding the training process and the model's performance. Can the authors also plot the learning curves and other metrics in the training?

3. How are the datasets sampled for each task? The current information is insufficient to determine whether the test datasets are fair. More details on the sampling process and how fairness is ensured are needed.

**Limitations:**

Yes.

---

> ### Author Rebuttal · Authors · 2024-08-07
>
> ### Weakness
> > **Q1(a). Generation of ICs of GS model for training and testing data.**
>
> First, to create ICs for the GS equation, we define a grid based on the spatiotemporal resolution and initialize the concentrations of chemicals A and B. Second, we set different random seeds to add random noise, changing the values of the chemicals at different positions. Finally, we obtain different ICs. Hence, the training and testing samples are independent. We will clarify this in our revised paper.
>
> > **Q1(b). Concern on generalization with limited training data.**
>
> Our model is designed to adhere to PDEs. While different ICs lead to varied steady-state patterns, these variations are ultimately governed by the underlying PDEs. Our model encodes the PDEs within a learnable PDE block which helps reduce the model’s dependence on data (please see **our reply to Q3 in General Reply**). This module ensures the model to accurately capture the underlying dynamics even on a coarse grid and with **limited data**. Since PDE is embedded into the network with RK4 integrator, the model is able to generalize prediction of solution trajectories over different ICs.
>
> Moreover, we employ different rollout timesteps during the training stage to improve the model's performance. That is, we treat a geneirc frame as an IC, and divide the trajectories into many samples based on the selected rollout timestep, which are then shuffled and grouped into batches for training (see Section 4.1 in our paper). As a result, the trained model is capable of extrapolating the prediction in the time horizon.
>
> Finally, we also demonstrated that our model can generalize to different types of boundaries (as discussed in **our reply to Q1 in General Reply**).
>
> > **Q2. Clarifications on $\tilde{x}$ and $x$, and $u$ in Eq. (1) and use $\hat{u}$ in Eq. (3).**
>
> We downsample the data from the fine grid to the coarse grid, denoted by $x$ and $\tilde{x}$ respectively, as mentioned in Sections 3.1 and 3.2.1 in our paper. We denote $\mathbf{u}$ as the ground truth of spatiotemporal dynamics and $\hat{\mathbf{u}}$ as the solution after passing through the correction block, resulting in a corrected coarse solution. This process can be represented as $\hat{\mathbf{u}}_k=\texttt{NN} (\mathbf{u}_k)$, where the correction block is a trainable network defined in Section 3.2.2. Hope this clarifies your question.
>
> > **Q3(a). Fair comparisons between our method and baselines.**
>
> Our baselines include both purely data-driven (UNet and FNO) and physics-aware (PeRCNN and LI) methods. The reason why we consider UNet and FNO as baselines is to test their extreme performance on small datasets. Initially, we also considered DeepONet, but it was removed from the NS case due to its poor performance in the first three experiments.
>
> We have also added temporal stencil modeling (TSM) [Sun, et al. ICML 2023] and learned correction (LC) [Kochkov, et al. PNAS 2021] as additional baselines, which use a history of solution as input and incorporate physics knowledge. The correlation curve is ploted in **Fig. R2** in the *1-page PDF rebuttal file*. We believe the additional experiments would improve the fairness in the baseline comparisons.
>
> > **Q3(b). Details of settings and configurations of runtine comparisons in Figure 6.**
>
> Please see **our detailed response to Q4 in General Reply**. We will clarify this in our revised paper.
>
> ### Questions
> > **Q1. The purpose and necessity of the lower path in Figure 1(a).**
>
> We present the role of each module in **our response to Q2 in General Reply** and discuss their importance in **our response to Q3 in General Reply**. Given the potential instability and accumulated errors in the learnable PDE block, especially in the NS example, the lower path is indispensable. It serves to correct the solution generated by the PDE block. Moreover, since the input of the NN block includes pressure ${p}$ to provide more information, the lower path also includes an optinal Poisson block. Adding such a block help improve the model's performance as shown in **Table R6** in the *1-page PDF rebuttal file*.
>
> > **Q2. Initialization of conv filters and the correction block, and learning curves in training.**
>
> We use random initialization for the symmetric Conv filters and scale the parameters to a low magnitude with an empirical value of 0.001. For the correction module, the Conv filters use the default Kaiming Uniform initialization, while other components use random initialization. We also plot the learning curves (averaged over 5 independent training trails for the NS dataset) as an example, shown in **Fig. R4** in the *1-page PDF rebuttal file*.
>
> > **Q3. Sampling process for datasets and fairness of the testing datasets.**
>
> For the Burgers, FN, and GS datasets, which utilize the finite difference method as solver, the solution is calculated at mesh grids. To establish a coarse grid ground truth, we apply uniform downsampling [Rao, et al. Nature Machine Intelligence 2023] in both temporal and spatial dimensions. In contrast, the NS dataset, which uses the finite volume method, computes solutions at cell faces in a staggered manner. Therefore, we use staggered average downsampling [Kochkov, et al. PNAS 2021] (shown in **Fig. R5** in the *1-page PDF rebuttal file*) in the spatial dimension and uniform downsampling in the temporal dimension.
>
> For fairness of testing datasets, we randomly select 10 seeds from the range of [1, 10,000], which are used to generate different Gaussian noise disturbances, which are then applied to the initial velocity field to create varying ICs. We also perform a warm-up phase during data generation to ensure that the variance and mean of the trajectories are closely aligned, thereby maintaining fairness. We wiil clarify this in the revised paper.
>
> **Remark:** Thanks for your constructive comments. Your consideration of improving the rating of our paper will be much appreciated!

---

> ### Author Response · Authors · 2024-08-10
> **Sincerely looking forward to your feedback**
>
> Dear Reviewer WfL9,
>
> We hope our point-by-point response in the rebuttal has addressed your concerns. We are very much looking forward to your feedback during the discussion period. We would be more than happy to answer any further questions you may have.
>
> Best regards,
>
> The Authors

---

> ### Author Response · Authors · 2024-08-11
> **Look forward to further discussions before the end of the discussion period**
>
> Dear Reviewer WfL9:
>
> As the author-reviewer discussion period will end soon, we would appreciate it if you could kindly review our responses at your earliest convenience. If there are any further questions or comments, we will do our best to address them before the discussion period ends.
>
> Thank you very much for your time and efforts!
>
> Sincerely,
>
> The Authors

---

> > ### Comment · Reviewer_WfL9 · 2024-08-12
> >
> > Thanks for your detailed reponse. I raised my score.

---

> > > ### Author Response · Authors · 2024-08-12
> > > **Thank you for raising the score**
> > >
> > > Thank you for your positive feedback and for increasing the score. We will include the additional experiments and text in the revised paper.

---

### Official Review · Reviewer_xwvp · 2024-07-10

**Soundness:** 4
**Presentation:** 3
**Contribution:** 3
**Rating:** 6
**Confidence:** 5

**Summary:**

In this paper the authors propose a PDE preserved coarse correction network for efficient prediction of spatio-temporal dynamics. The aim is to develop a learnable coarse model that accelerates simulation and prediction of spatio-temporal dynamics based on down-sampled data. The method mainly consists of 4 blocks, PDE block for computing spatial derivatives on coarse grids, correction block, NN block and Poisson block. The method proposed by the authors operates in tandem with the RK-4 numerical integration scheme and uses spectral methods to estimate the pressure field on the fly to condition the networks for predicting better corrections. The method yields significant accuracy gains over baselines for several use cases and showcases impressive generalization.

**Strengths:**

- Trainable conv filter with symmetric constraints
- Periodic BC padding embeds physical constraints.
- Poisson block computes the pressure field on the fly using a spectral method.
- The integration of ML methods with the numerical time integrators such as RK-4 to develop robust end-to-end solutions
- Generalization and accuracy gains are very impressive over long time horizons, considering the fact that limited ICs are used for training.

**Weaknesses:**

- Is the method only applicable for problems with periodic BCs? The effectiveness of the derivative calculation as well as the imposition of the BCs will need to be verified.
- Require a structured grid for the method to work as subsampling can be very difficult for unstructured methods especially when non-uniform input conditions such as geometry or source terms are involved. Additionally, developing a kernel for spatial derivative calculation might not be trivial, application of BCs and solving for the pressure on the fly will be challenging.
- Method needs to be explained in a better way. Some of the questions related to the method are outlined below.

**Questions:**

- It seems like the Poisson block, NN block are dependent on the PDE block. If I consider the sequence of actions then the PDE block needs to be evaluated first. Fig 1 needs to be modified to reflect that.
- In the PDE block, 3 types of inputs go to the RK-4 scheme, original solution, corrected solution and then the output of the poisson block. It is not clear how these different components are used for time integration? What had to be changed in the PDE block to implement the Euler scheme?
- How is the Poisson block stable when initially the velocity derivatives calculated by the untrained weights are inaccurate? Wouldn’t this cause severe instability early on in the training? Is there some under-relaxation required?
- It needs to be clarified that the equation residue block is basically the RK-4 integration.
- Authors claim that “although the number of learnable parameters are limited, the coarse derivatives can still be accurately approximated after the model is trained (see ablation study in 4.2)”. However this claim cannot be verified from the ablation study that the derivatives computed truly match the derivatives computed at the same location from a finer grid. In order to verify this, the authors will need to compare the gradient computed by their kernel at a certain location on the coarse grid and compare it to the finite-difference gradients at the same location from a fine grid at the same snapshot in time.
- There is a slight mismatch in the methodology between the baselines and the proposed approach. The baselines are designed to learn the mapping to predict the next solution, whereas the proposed method is designed to learn the correction. The paper should ideally include some baselines that are trained to learn just the correction as your method so that true advantage of the proposed approach is clear? Additionally, most of these baselines perform better when a history of solutions is provided as an input. Were the baselines such as FNO and UNet trained with a time history?
- Why is a filter of 5x5 enough? Should the size of the filter be dependent on the amount of coarsening? Is the 4th order accuracy of the derivatives verified? How are the gradients calculated at the boundaries for non-periodic BCs?
- Authors claim that the learnable PDE block is enough in many scenarios, can this be verified through an ablation study? One of the ablation studies should involve removing the NN block to verify that the gradients predicted by the PDE block truly contribute to the accuracy gains achieved by the method.
- Another ablation study that would be interesting is to sum up the derivative from finite-difference kernel and from the trainable conv symmetric filter in the PDE block and compare it with just the derivative from the trainable conv filter.
- Ablation study to understand the importance of the Poisson block is also required. Is the Poisson block really required? What happens if the pressure is also predicted as an output of the network? How would those results compare.
- I don’t agree with the authors claim that the time marching in the network will inherently generalize the predictions to all ICs. Small discretization errors due to coarse grids can accumulate and the solution trajectories can be significantly different. The burden is actually on the network predicting corrections to correct the deviation in the trajectory.
- How different are the training and testing initial conditions?
- Need to see correlation plots over time steps for all the use cases to show how much the prediction and ground truth decorrelate over time for the proposed method as well as baselines.
- Computational cost of the method?

---

> ### Author Rebuttal · Authors · 2024-08-07
>
> ### Weakness
> > **W1. Different BC encoding and derivative calculations.**
>
> Despite we showcase the eficacy of our model on datasets with Periodic BCs, it is applicable to handle other types of BCs (see **Table R2**, **Table R5** and **Fig. R1** in the *1-page PDF rebuttal file*). **Please see our detailed reply to Q1 in General Reply**.
>
> > **W2. Challenges for unstructured grids.**
>
> Great remark! We acknowledge the challenges on extending the current pipeline to unstructured meshes. Nevertheless, the overarching framework is extensible, e.g., integrating graph neural networks and differentiable finite element/volume methods to learn on unstructured meshes. We will discuss the challenges in the revised paper.
>
> ### Questions
> > **Q1. Re-organization of PDE-block, Poisson block, & NN block in Fig. 1.**
>
> Considering the sequence of actions, placing the PDE block in Fig. 1.b first, followed by the Poisson block, would be better. We will modify it in the revised paper.
>
> > **Q2. Clarification of three input components for time integration in the PDE block.**
>
> Please see **our reponse to Q1 in General Reply**.
>
> > **Q3. Stability of Poisson block in early stage of training.**
>
> The Poisson block mainly relies on a numerical solver without any trainable parameters. We scale the initialized weights of the symmetric filter to ensure that the Laplacian term has a low magnitude. Thanks to the correction block, the solution does not reach a large magnitude, either. We added a loss curve to prove this point (averaged over 5 independent training trails), shown in **Fig. R4** of the *1-page PDF rebuttal file*. The consistent performance across these trials demonstrates that our model does not encounter instability during early training. We also explored relaxing the Poisson block by adding trainanle parameters (e.g., scaling coefficients). However, no improvement is observed.
>
> > **Q4. Equation residue block is the RK-4 integration.**
>
> You are right -- the PDE block is indeed designed based on the RK4 integrator. However, the PDE block is trainable, consisting of the correction block and a trainable filter bank, which differs from the standard RK4-based finite diference (FD) solver.
>
> > **Q5. Comparing derivatives from kernels on coarse grid with those from FD on fine grid.**
>
> The derivatives of these two strategies are different. We learn an approximate surrogate of derivatives on the coarse grid, which are different from the ground truth derivative values. Our goal is to minimize the overall PDE residual, instead of making each derivative equal to the corresponding ground truth.
>
> > **Q6. More baselines that learn corrections and take history of solutions as inputs.**
>
> We further considered 3 correction learning baselines, including learned correction (LC) [PNAS 2021, 118(21):e2101784118], two variants of UNet and FNO by adding the NN Block for correction. The results in **Table R1**, in the *1-page PDF rebuttal file*, showed that our model has the best performance.
>
> We have also trained UNet and FNO that use 12 history snapshots as inputs. However, their performance didn't improve. Note that LI and TSM in our baselines also use historical information (32 steps) as inputs. These baselines cannot capture the underlying dynamics due to the limited training samples.
>
> > **Q7. Filter size and its dependence on grid size.**
>
> Following your suggestion, we conducted experiments on different filter sizes, e.g., 3x3, 5x5, 7x7. As shown in **Table R4** in the *1-page PDF rebuttal file*, the model with the 5x5 filter outperforms the other two (the 3x3 filter has low accuracy while the 7x7 filter has instability issue on coarse grids). Hence, the 5x5 filter is empirically suggested.
>
> > **Q8. Ablation on removing the NN block.**
>
> For simple systems, removing the NN block does not affect the results. For complex systems, e.g., NS, it causes the long rollout prediction diverged. Please see **our reply to Q3 in General Reply**.
>
> > **Q9. Ablation to replace trainable symmetric filter with FD kernel in the PDE block.**
>
> We conducted an ablation study on the Burgers dataset, e.g., replacing our trainable conv filter with FD kernel in the PDE block (NN block excluded in the model). However, this led to NaN values in the test results due to large error accumulation. This indicates that FD kernels fails to approximate the derivatives on coarse grids.
>
> > **Q10. Ablation on the Poisson block.**
>
> We conducted an ablation study on the NS dataset to evaluate the significance of the Poisson block, as shown in **Table R6** in the *1-page PDF rebuttal file*. We can see that the model's performance deteriorates without the Poisson block.
>
> > **Q11. Generalization to all ICs; discretization error accumulation.**
>
> Since PDE is embedded into the network with RK4 integrator, the model can in theory generalize prediction of solution trajectories over different ICs. However, we agree with you that large error accumulation over time might occur given a complex discretized IC on coarse grids. We will clarify this in our revised paper.
>
> > **Q12. Difference between the training and testing ICs?**
>
> We use random seeds to generate different ICs. In the NS example, different ICs are created by generating random noise for each component of the velocity field and filtering it to create a divergence-free field with desired properties. We randomly selected 10 ICs and drew histograms to show the differences, shown in **Fig. R3** in the *1-page PDF rebuttal file*.
>
> > **Q13. Correlation plots over time steps.**
>
> The correlation curve over time steps for the NS dataset is shown in **Fig. R2** in the *1-page PDF rebuttal file*. To provide a more comprehensive comparison, we also included DNS results for various grid resolutions in the figure.
>
> > **Q14. Computational cost.**
>
> Please see **our reply to Q4 in General Reply**.
>
> **Remark:** Thanks for your constructive comments. Your consideration of improving the rating of our paper will be much appreciated!

---

> ### Author Response · Authors · 2024-08-10
> **Sincerely looking forward to your feedback**
>
> Dear Reviewer xwvp,
>
> We hope our point-by-point response in the rebuttal has addressed your concerns. We are very much looking forward to your feedback during the discussion period. We would be more than happy to answer any further questions you may have.
>
> Best regards,
> The Authors

---

> ### Author Response · Authors · 2024-08-11
> **Look forward to further discussions before the end of the discussion period**
>
> Dear Reviewer xwvp:
>
> As the author-reviewer discussion period will end soon, we would appreciate it if you could kindly review our responses at your earliest convenience. If there are any further questions or comments, we will do our best to address them before the discussion period ends.
>
> Thank you very much for your time and efforts!
>
> Sincerely,
>
> The Authors

---

### Author Rebuttal · Authors · 2024-08-07

## General reply

We deeply appreciate the insightful and constructive comments from the reviewers, which are helpful in improving our paper. We are pleased that all the reviewers recognized the novelty and excellent generalizability of our work. In particular, we thank the reviewers for recognizing the *robustness* (xwvp and aBLf), *interpretability* (WfL9), and *data-efficiency* (bvnV) of our method.

We have summarized a detailed reply to several **common questions** and addressed other concerns in each individual rebuttal. In addition, we have also listed **six tables** and **five figures** in the *1-page PDF rebuttal file* to support our rebuttal.

> **Q1. Applicability to different BCs beyond periodic BCs.**

Excellent comment! To verify the applicability of our model to different BCs, we use the Burgers equation as an example and set the left boundary as *Dirichlet*, the right boundary as *Neumann*, and the top/bottom boundaries as *Periodic*. Here, we denote this case as Complex Boundary Conditions (CBC). The rest of the data generation setup (e.g., ICs, mesh grids) remains the same as used in the paper. We generated 10 CBC test datasets resulting from different random ICs.

We then directly tested the model previously trained based on Periodic BC datasets reported in the paper, meanwhile processing the boundaries using the BC encoding strategy as shown in **Table R2** in the *1-page PDF rebuttal file* during inference. The quantitative results (average over 10 datasets) are presented in **Table R5** in the *1-page PDF rebuttal file*, where we also list the predicted snapshots at 1.4 s for two random ICs in **Figure R1**. We can see that our model is capable of generalizing over different BCs. We will add this result in the revised paper.

> **Q2. The flow of data between blocks and their explanation.**

In Figure 1a, the network architecture includes two paths: the upper path computes the coarse solution using a learnable PDE block, while the lower path is incorporated into the network to correct the solution on a coarse grid with a Poisson block and a NN block. The data flow operates as follows.

-The network accepts $\mathbf{u}_k$ as input and processes it by the PDE block on the upper path, where the PDE block computes the residual of the governing equation. A filter bank, defined as a learnable filter with symmetry constraints, calculates the derivative terms based on the corrected solution (produced by the correction block). These terms are combined into an algebraic equation (a learnable form of $\mathcal{F}$). This process is incorporated into the RK4 integrator for solution update.

-In the lower path, ${\mathbf{u}_k}$ is first corrected by the correction block, and ${p_k}$ is computed by the Poisson block. Inputs, including solution states $\{\mathbf{u}_k, p_k\}$ and their derivative terms, forcing term, and Reynolds number, are fed into the NN block. The output from this block serves as a correction for the upper path.

-The final result $\mathbf{u} _{k+1}$ is obtained by combining the outputs from both the upper and lower paths. During the gradient back-propagation process, NN block learns to correct the coarse solution output of the PDE block on the fly, and ensures that their combined results more closely approximate the ground truth solution.

We will include more details in the revised paper to make this procedure more straightforward.

> **Q3. The importance of PDE block, NN block, and Poisson block.**

We found that using a simplified version of the model (without the parallel NN block) can still achieve SOTA performance on relatively simple systems such as the Burgers, FN, and GS datasets. That is said, the learnable PDE block itself alone has satisfactory performance. However, for the NS equation, given the complexity of the flow ($Re=1000$), we need the NN block as a supplement to alleviate the error accumulation of long-term prediction and the instability issue on coarse grids.

In fact, the learnable PDE block aims to reduce the network's reliance on training data. To achieve better results, both the PDE block and NN block are necessary. We clarify that the Poisson Solver is a numerical solver without learnable parameters. Moreover, we conducted an additional ablation study via removing (1) the PDE block and (2) the Poisson block, respectively. These results are shown in **Table R6** in the *one-page PDF rebuttal file*, and will be added to the revised paper. We can see that both blocks are essential to maintain satisfactory prediction accuracy.

> **Q4. Training and inference time.**

We recorded the training and inference time taken by the model on Burgers, FN, GS, and NS in **Table A** (with the inference time steps of 1400, 1250, 1000, and 1200, respectively). Please note that this result is based on optimized codes showing a better performance compared with that reported in Figure 6 in the paper (this figure will be updated). It is observed that the training time of our model on the first three datasets are relatively short. On the NS dataset, due to the increased model complexity, the training time increases accordingly. Nevertheless, the inference time for all cases remians similar. Note that both training and inference are performed on a single 80GB Nvidia A100 GPU, as described in Appendix F of the paper.

**Table A.** The effiency of P$^2$C$^2$Net on different datasets.

| Time cost | Burgers | FN | GS | NS |
| -------- | -------- | -------- | -------- |-------- |
|  Training   | 2h     |  0.6h   |  2.5h    | 69h  |
|  Infer Cost   | 17s     |  16s    |  15s    | 20s  |

Please do feel free to let us know if you have any further questions. Thank you very much!

Best regards,

The Authors of the Paper

---

### Decision · Program_Chairs · 2024-09-25

**Decision:**

Accept (poster)

**Comment:**

The paper proposes a new correction method for solve PDEs on coarse meshes. The reported numerical results are quite convincing, giving significant improvement. All the reviewers agree that the paper should be accepted, and I agree as well.